# Global large-scale stratosphere-troposphere exchange in modern reanalyses

Alexander C. Boothe[1] and Cameron R. Homeyer[1]

[1]School of Meteorology, University of Oklahoma, Norman, Oklahoma, USA.

*Correspondence to:* Alexander C. Boothe (alexboothe92@ou.edu)

**Abstract.** Stratosphere-troposphere exchange (STE) has important impacts on the chemical and radiative properties of the upper troposphere and lower stratosphere. This study presents a 15-year climatology of global large-scale STE from four modern reanalyses: ERA-Interim, JRA-55, MERRA-2, and MERRA. STE is separated into three regions (tropics, subtropics, and extratropics) and two transport directions (stratosphere-to-troposphere transport or STT and troposphere-to-stratosphere transport or TST) in an attempt to identify the significance of known transport mechanisms. The extratropics and tropics are separated by the tropopause "break". Any STE occurring between the tropics and the extratropics through the tropopause break is considered subtropical exchange (i.e., in the vicinity of the subtropical jet).

In addition, this study employs a method to identify STE as that which crosses the lapse-rate tropopause (LRT), while most previous studies have used a potential vorticity (PV) isosurface as the troposphere-stratosphere boundary. PV-based and LRT-based STE climatologies are compared using the ERA-Interim reanalysis output. The comparison reveals quantitative and qualitative differences, particularly for TST in the polar regions.

Based upon spatiotemporal integrations, we find STE to be STT-dominant in ERA-Interim and JRA-55 and TST-dominant in MERRA and MERRA-2. The sources of the differences are mainly attributed to inconsistencies in the representation of STE in the subtropics and extratropics. Time series during the 15-year analysis period show long-term changes that are argued to correspond with changes in the Brewer-Dobson circulation.

## 1 Introduction

Stratosphere-troposphere exchange (STE) has significant impacts on the chemical and radiative properties of the upper troposphere and lower stratosphere (UTLS; Holton et al., 1995). Oxidative and greenhouse gases can be transported across the tropopause in two directions, typically referred to as stratosphere-to-troposphere transport (STT) and troposphere-to-stratosphere transport (TST). STT brings ozone-rich stratospheric air into the troposphere, and in some cases STT can extend into the planetary boundary layer (e.g., Danielsen, 1968; Lin et al., 2012). Per molecule, ozone radiative forcing is maximized in the upper troposphere (Lacis et al., 1990). Conversely, TST processes can inject water vapor and tropospheric pollutants, such as carbon monoxide, into the lower stratosphere, where the lifetimes of such gases can be increased. Because water vapor is a greenhouse gas, increases in LS water vapor from TST lead to an increase in radiative forcing similar to that for UT ozone (Forster and Shine, 2002).

STE is driven by dynamic processes occurring across a wide range of spatial and temporal scales. There are several known large-scale processes that occur in the extratropical domain, the tropical domain, and along the boundary between them (i.e., within the subtropics). STE in the extratropics often occurs in the vicinity of baroclinic transient eddies, or extratropical cyclones (e.g., Holton et al., 1995; Wernli and Davies, 1997; Wirth and Egger, 1999; Reutter et al., 2015). Transport associated with extratropical cyclones is dominated by STT, and is primarily due to clear-air turbulence along the edges of stratospheric intrusions (or tropopause folds) in the upper-troposphere (e.g., Danielsen, 1968; Shapiro, 1980; Lamarque and Hess, 1994; Cooper et al., 2004; Reutter et al., 2015). Stratospheric intrusions can develop apart from extratropical cyclones along the cyclonic side of upper tropospheric jet streams as a result of ageostrophic circulations (e.g., Sawyer, 1956; Shapiro, 1981; Shapiro and Kennedy, 1981; Keyser and Shapiro, 1986). These tend to be shallow and exchange less than stratospheric intrusions associated with extratropical cyclones (e.g., Wernli and Bourqui, 2002; Sprenger and Wernli, 2003). Extratropical cyclones also result in TST, which is sourced by warm-conveyor-belt flows that bring lower troposphere air to the UTLS along isentropic surfaces and by moist convection (e.g., Stohl, 2001; Wernli and Bourqui, 2002; Reutter et al., 2015). Similar to transient extratropical cyclones, cut-off extratropical cyclones (i.e., limited interaction with middle latitude westerlies) occur frequently in the upper-troposphere and result in STT due to eroding processes (i.e., turbulence and friction) acting upon the depression in the tropopause (Price and Vaughan, 1993).

STE has been found to be globally unbalanced, with net TST in the tropics and net STT in the extratropics. This imbalance is primarily a reflection of ubiquitous tropospheric upwelling at the tropical tropopause and downwelling at the extratropical tropopause that is driven by the Brewer-Dobson circulation (or BDC; Brewer, 1949; Dobson, 1956). The BDC is a latitudinal circulation, where tropospheric air diabatically ascends into the tropical lower-stratosphere and is dynamically "pumped" poleward and downward into the extratropical LS by Rossby wave breaking in the mid-latitudes (Holton et al., 1995). Apart from the BDC, TST in monsoon anticyclones is an important STE mechanism in the tropics. Moist convection rapidly lofts lower troposphere air into the interior of the UT anticyclonic circulation where it slowly ascends into the lower stratosphere. The North American and Asian Monsoon Anticyclones have received a considerable amount of recent attention and have been shown to contribute significantly to global STE (e.g., Randel et al., 2010, and references therein).

There are also known STE processes that occur primarily in the subtropics and involve tropical UT air and extratropical LS air. One of the most well-known subtropical STE processes is Rossby wave breaking, which is a quasi-isentropic transport process that can be predominantly poleward (TST), equatorward (STT), or bidirectional (e.g., Seo and Bowman, 2001; Scott and Cammas, 2002; Homeyer and Bowman, 2013). Stratospheric intrusions along the subtropical jet stream occur outside of the tropics, but can bring extratropical lower stratosphere air into the tropical middle and upper troposphere (Waugh and Polvani, 2000).

The impacts of STE are considerable on the global scale but, as a result of limited observations, most studies have investigated STE and associated large-scale processes over small domains and for short time periods using numerical models and remote sensing or aircraft platforms. For example, Lamarque and Hess (1994) used a simulated stratospheric intrusion to estimate the annual net mass exchange across the tropopause in the Northern Hemisphere by extrapolation. Specifically, annual estimates of STT were approximated using an annual average of stratospheric intrusions (1460 per year) and STT quantities

found in their case study, which amounted to $1.79 \times 10^{17}$ kg per year. Although quantitative estimates from Lamarque and Hess (1994) were similar to additional studies published at the time, they suggested that stratospheric intrusions are invariable and exhibit similar lifespans and STE quantities, which is now known to be an incorrect assumption. Modern numerical models and computing capabilities, however, enable climatological studies of STE for large domains and long time periods.

Previous studies have employed multiple years of global model output to produce a climatology of STE. Appenzeller et al. (1996) analyzed downward fluxes out of the lower stratosphere in the Northern and Southern Hemispheres over two calendar years (1992 and 1993). The study emphasized an Eulerian approach, the "downward control principle" (i.e., mass continuity), to estimate seasonal net flux in the extratropics, and results indicated a peak downward flux during Northern Hemisphere spring. Annual net downward mass flux was estimated to be $3.5 \times 10^{17}$ kg and $3.3 \times 10^{17}$ kg for the Northern and Southern hemispheres,

respectively, which is nearly twice that found by Lamarque and Hess (1994). An important caveat in the Appenzeller et al. (1996) study is that the method does not provide spatial distributions of STE and only considers STT in the extratropics. Another Eulerian metric used as a diagnostic for STE is the Wei method (Wei, 1987), which allows for determination of STT and TST on short spatiotemporal scales but suffers from conceptual problems (Gettelman and Sobel, 2000).

An alternative to Eulerian methods is a Lagrangian approach, which employs a large number of three-dimensional (or 3-

D) trajectories to determine STE. For example, Stohl (2001) developed a one-year Northern Hemisphere climatology of STE using trajectories and identified large-scale airstreams and the corresponding spatial and temporal variability of STE associated with particular flows (i.e., warm-conveyor belts and stratospheric intrusions). Annual net downward mass flux estimates in the northern extratropics were found to be larger than previous global estimates ($4.4 \times 10^{17}$ kg). Multi-year climatologies of STE have also employed a Lagrangian approach on global and hemispheric scales, wherein studies aim to investigate

the quantitative and qualitative characteristics of STE (e.g., Wernli and Bourqui, 2002; Sprenger and Wernli, 2003; Škerlak et al., 2014). A recent STE climatology by Škerlak et al. (2014, hereafter Š14) evaluated STE over a 33-year period using 3-D kinematic winds from the ERA-Interim reanalysis and a Lagrangian trajectory model. Their method requires a parcel to remain in its parent reservoir (troposphere or stratosphere) and destination reservoir (stratosphere or troposphere) for at least 48 hours to be classified as irreversible STE (a so-called "residence time"). Although a Lagrangian trajectory method was employed, the

annual net exchange found ($0.42 \times 10^{17}$ kg) was an order of magnitude smaller than estimates by Stohl (2001). It is apparent that residence times significantly decrease the quantitative STE estimates as a result of separating irreversible exchange from transient exchange.

As outlined above, there have been three classes of methods used to analyze STE: (1) Eulerian based approaches, such as the Wei method and the "downward control principle", (2) a Lagrangian trajectory-based method that does not use a residence filter

(e.g., Stohl, 2001; Seo and Bowman, 2002), and (3) a Lagrangian trajectory-based method with a residence filter (e.g., Wernli and Bourqui, 2002; Sprenger and Wernli, 2003; Škerlak et al., 2014). Despite previous efforts to determine climatological characteristics of STE, the various models, methods, and time periods used have led to a wide range of transport estimates.

Motivated by an improved understanding of UTLS characteristics and processes and the availability of output from multiple higher-resolution global models, this study seeks to develop and contrast climatological estimates of STE from modern global

reanalyses: the European Centre for Medium-range Weather Forecasting Interim reanalysis (ERA-Interim), the Japanese Me-

teorological Agency 55-year reanalysis (JRA-55), and the National Aeronautics and Space Administration (NASA) Modern Era Retrospective analysis for Research and Applications (MERRA) and version-2 of this system (MERRA-2). We apply a Lagrangian approach using 3-D kinematic wind fields from each reanalysis to compute STE during a 15-year period: 1996-2010. STE is further separated into three regions in an attempt to evaluate the role of well-known large-scale transport processes on the global scale: tropical, subtropical, and extratropical. The direction of transport (TST or STT) is also separately evaluated for each region. While small-scale mechanisms including gravity wave breaking and convection are also known to contribute to STE, such processes are not investigated in this study because they are not resolved in the reanalyses.

## 2   Data and methods

### 2.1   Reanalysis model output

As outlined in Section 1, we employ output from four reanalysis models in this study: ERA-Interim, JRA-55, MERRA, and MERRA-2. ERA-Interim is available from 1979–present on an approximately 80 km horizontal grid and at 60 vertical model levels with a model top of 0.1 hPa (Dee et al., 2011). JRA-55 is available from 1958–present on a ∼60 km horizontal grid (Kobayashi et al., 2015). Similar to ERA-Interim, JRA-55 has 60 vertical model levels with a model top of 0.1 hPa. MERRA is available from 1979–2016 at $0.5° \times 0.667°$ horizontal resolution and at 72 vertical model levels with a model top of 0.01 hPa (Rienecker et al., 2011). MERRA-2 has a similar design to MERRA, but is available at a slightly finer horizontal resolution of $0.5° \times 0.625°$ from 1979–present (Bosilovich et al., 2015). Although the horizontal and vertical resolution is similar in MERRA and MERRA-2, numerical improvements in MERRA-2 are expected to better represent UTLS processes. We use both MERRA and MERRA-2 to show improvements with respect to UTLS processes, specifically STE. For a more detailed discussion of reanalyses and their differences, see Fujiwara et al. (2017).

Furthermore, we employ 3-D output at 6-hour intervals from each reanalysis in this study. All of the data is interpolated to a regular $1° \times 1°$ latitude-longitude grid for analysis. Model output is also used to calculate secondary variables for analysis: tropopause pressure (using the World Meteorological Organization lapse-rate tropopause (LRT) defined by WMO (1957)), potential temperature ($\theta$), and potential vorticity (PV) on the native model levels (i.e., no vertical interpolation is performed). It is important to note that, while the interpolation to a regular horizontal grid acts to slightly coarsen the original data, the interpolation has no measurable effect on the LRT altitude distribution and, given the 6-hour time resolution, is expected to have little impact on trajectory model solutions (e.g., see Stohl, 1998, and references therein).

### 2.2   Tropopause definition

The tropopause definition employed in STE studies is critical to their outcome since it represents the boundary between troposphere and stratosphere and the location where dynamical processes and associated transport are evaluated. Many previous

STE studies have used a "dynamical" tropopause to represent the troposphere-stratosphere boundary, for which a potential vorticity (PV) isosurface such as 2-PVU (1 PVU = $10^{-6}$ Km$^2$kg$^{-1}$s$^{-1}$) is used (Ertel and Rossby, 1949). However, several studies demonstrate that STE estimates can be largely sensitive to small changes in the PV isosurface used. For example, Seo and Bowman (2002, their Fig. 6) made Lagrangian estimates of STE using multiple control surfaces, including isobaric surfaces and potential vorticity iso-surfaces, and found downward mass flux ranged from $1 - 4 \times 10^{17}$ kg/yr. Homeyer and Bowman (2013) used 30 yr of ERA-Interim to produce a climatology of Rossby wave breaking events and associated STE in the subtropics and demonstrated that varying the PV boundary from 2 to 4 PVU resulted in a reversal of the net transport direction (i.e., TST or STT). Despite these known sensitivities, many studies have continued to employ a dynamic tropopause to avoid challenging STE calculations in the vicinity of the sharp LRT discontinuity near the subtropical jet known as the "tropopause break" (e.g., Palmén, 1948; Randel et al., 2007; Homeyer and Bowman, 2013). Because PV is a quasi-conserved quantity in an adiabatic and frictionless flow, it is treated as a quasi-material surface and can provide a continuous boundary through the tropopause break. An additional aspect of a PV-based method that is problematic for global STE studies is the fact that PV values converge to zero at the equator, such that an isosurface does not coincide with the altitude of the tropical tropopause south of the tropopause break. As a result, most studies that use a PV-based method in the extratropics select an alternative surface to represent the tropopause in the tropics (e.g., cold point altitude, isentropic surface).

The LRT, on the other hand, has been shown to commonly coincide with the sharpest stability and chemical transitions between the troposphere and stratosphere globally (e.g., Gettelman et al., 2011; Pan et al., 2004). This is due to the fact that a single PV value does not coincide with the LRT and chemical transition everywhere, with PV values at the tropopause ranging from at least 1 to 6 PVU in the extratropics. As a result, some recent studies advocate for and employ a PV gradient-based tropopause for STE studies (e.g., Kunz et al., 2011, 2015), but such an approach continues to suffer from an inability to represent the troposphere-stratosphere transition in the tropics. One exception to the success of the LRT as a reliable troposphere-stratosphere boundary is in the Antarctic region during Austral winter, where reduced stability in the upper troposphere leads to an erroneously high LRT altitude (Zängl and Hoinka, 2001). However, this issue is limited to latitudes poleward of 60°S ($\sim$6% of Earth's area) and about 3 months out of the year. For these reasons, we use the LRT to represent the global troposphere-stratosphere boundary in this study, but leverage beneficial information from PV analyses to identify irreversible transport. A detailed outline of this approach is provided in the following subsection. Apart from infrequent biases in the LRT altitude, its uncertainty is comparable to the UTLS vertical resolution of the dataset used, which varies amongst the reanalysis models. An approximate altitude range of the UTLS is 8-14 km in the extratropics and 13-18 km in the tropics. Vertical resolution in MERRA and MERRA-2 is about 1 km throughout the extratropical and tropical UTLS, whereas JRA-55 and ERA-Interim vertical resolution ranges from about 750 m in the extratropical UTLS to about 1100 m in the tropical UTLS. These differences in resolution may contribute to differences in STE found between the reanalyses outlined in the remainder of the paper.

## 2.3 STE identification

Trajectory calculations ($\sim 6 \times 10^9$) in this study are performed using the TRAJ3D model developed by Bowman (1993) and updated in Bowman and Carrie (2002). Parcels are initialized daily at 00 UTC every $1°$ in longitude and latitude and every 20 hPa at altitudes relative to the LRT. Analogous to STE methods in previous studies, preliminary selection of STE parcels is
5 dependent upon whether trajectories cross the LRT within the initial 24 hours downstream. This selection is a straightforward process and only requires the initial and final parcel pressure and coincident tropopause pressures. For instance, if a parcel pressure is initially lower than its coincident tropopause pressure and one day downstream the parcel pressure is greater than its coincident tropopause pressure, it is flagged as possible STT (regardless of the geometric evolution of the parcel). All potential STE parcels are then advected 5 days forward and backward in time from the initial parcel locations for further
analysis.

To ensure that transient (intermittent tropopause crossing) STE parcels are not erroneously counted and represented as irreversible exchange, a filtering method is applied. Two criteria are necessary to identify irreversible transport: (i) a residence time ($\tau$), and (ii) a parcel PV change occurring during the 10-day trajectory period. Based on a sensitivity study by Wernli and Bourqui (2002), a long residence time, longer than 24 hours, can decrease estimates of irreversible transport and change
the direction of the annual net STE. Also, a 4-day residence time allows STE processes with longer transport timescales to be identified. Here, we chose a strict residence time criteria, $\tau$, of 96 hours. The second filtering criteria requires an absolute PV change of 0.5 PVU from the initial parcel value to that 5 days downstream. The PV criteria represents a dynamic change in a parcel's characteristics from the influence of diabatic or frictional effects (i.e., mixing). Parcels that meet the required criteria are retained as irreversible exchange.
To examine irreversible STE mass flux, we compute the mass of each parcel based on the following equation:

$$M = \frac{1}{g}(a_0^2 \cos\phi)\Delta\lambda\Delta\phi\Delta p \tag{1}$$

where Earth's gravitational acceleration and radius are denoted by $g$ and $a_0$, respectively, and $\phi$, $\lambda$, and $p$ represent latitude, longitude, and pressure scales of each parcel. Since the parcel resolution is constant, parcel mass decreases from equator to pole. Therefore, a greater number of transported parcels are required in the extratropical and polar latitudes to achieve equiv-
25 alent STE to that in the tropics and subtropics. We bin STE parcels on a global grid with a longitude–latitude resolution of 2 degrees for analysis. For this purpose, the 1-day downstream parcel locations are used in an attempt to better represent the locations where STE occurred. Slight differences in the locations of STE in the comparisons with Š14 in Section 3 below may be due to this choice, since Š14 use their initial parcel locations for binning.

## 2.4 Categorizing STE

In an attempt to evaluate the role of individual large-scale STE processes globally, we separate transport into three regional categories: tropics, subtropics (i.e. between the tropical UT and extratropical LS), and extratropics. STT and TST occur in

each of the three regions and are counted separately. For large-scale STE processes, exchanges in the subtropics are known to correspond primarily with Rossby wave breaking, which is a quasi-horizontal (or isentropic) process. Exchanges in the extratropics and tropics, however, are associated with other processes, such as extratropical cyclones, stratospheric intrusions, downwelling in the extratropics and upwelling in the tropics.

We classify regions as tropical if they lie equatorward of the 'tropopause break' and extratropical if they lie poleward of the break. Similar to previous studies, the tropopause break is defined as the LRT altitude frequency minimum between tropical (15-17 km or $< 150$ hPa) and extratropical (8-12 km or $> 150$ hPa) modes in global and hemispheric distributions (e.g., Birner, 2010; Homeyer and Bowman, 2013, and Fig. 17 here). Analysis of tropopause altitudes in each reanalysis model reveals that, despite slight differences in LRT altitudes, the tropopause break can be routinely identified in each using a tropopause pressure threshold of 150 hPa. Therefore, we set trajectories with STE as tropical if the tropopause pressure at the initial parcel location is less than 150 hPa, and extratropical otherwise.

Similarly, classifying subtropical STE involves evaluating both its tropopause break-relative location and tropopause-relative altitude during advection. In particular, if a parcel is initially located below the tropical tropopause and located above the extratropical tropopause 5 days downstream, it will be flagged at as subtropical TST (i.e., exchanged poleward thorough the tropopause break). Alternatively, if a parcel is initially located above the extratropical tropopause and located below the tropical tropopause 5 days downstream, it is flagged as subtropical STT (i.e., exchanged equatorward through the tropopause break). Remaining parcels are flagged as either exchange occurring only in the extratropics or tropics. There is one important exception that must be accounted for: stratospheric intrusions below the subtropical jet. In order to identify these parcels as STE in the extratropics and not in the subtropics, we require an additional condition for STT in the subtropics: both initial and final parcel pressures must be less than (above) the initial extratropical tropopause pressure. See Fig. 1 for a detailed schematic of the identification method.

## 3  STE method comparison

As previously mentioned, there can be significant variability in climatological STE estimates based on the tropopause definition chosen to analyze exchange. Here, we use our STE geographic distributions from ERA-Interim to briefly illustrate some of the differences and similarities between our approach (i.e., using the LRT) and the recent STE climatology presented by Š14 that employs a dynamic tropopause (i.e., a PV isosurface of $\pm$ 2-PVU) in the extratropics and the 380 K isentrope in the tropics. In Figure 2, the global summertime and wintertime geographic distributions of STT are shown for each approach. During both seasons, the locations of STT maxima and minima are largely similar between the methods, but the magnitude of STT mass flux is significantly larger using the PV-based tropopause definition, particularly in the extratropics.

On the other hand, TST geographic distributions (Fig. 3) are shown to be quantitatively and qualitatively different. Similar to the STT comparisons, PV-based TST is larger than that found with the LRT method in most places. However, there is a unique latitudinal dependence of the differences, with the largest differences found in the polar regions of both hemispheres. For the

LRT method, global TST maxima are found within the tropics (i.e., monsoon anticyclones and ubiquitous tropical upwelling), while TST in the extratropics and especially the polar regions is found to be comparatively weak. The opposite relationship is found using a dynamic tropopause. According to our knowledge of large-scale processes associated with TST, the geographic distributions from the LRT-based climatology (left column Fig. 3) agree more closely with known transport mechanisms (e.g.,

5    Holton et al., 1995; Stohl et al., 2003; Sprenger et al., 2007).

The differences between the two methods may be rooted in the altitude placement of the tropopause definition used. The $\pm$ 2–PVU surface (dynamic tropopause used in Š14), while at times may reside at similar altitudes compared to the LRT, often resides at lower altitudes with respect to the LRT (e.g., Pan et al., 2004; Gettelman et al., 2011; Kunz et al., 2011). Therefore, UT stirring or mixing may lead to erroneously flagged STT or TST parcels when using the dynamic tropopause.

## 4    Results

Throughout the following section comparisons of STE among the four reanalysis systems are shown using various metrics. Here, we seek to reveal important similarities and differences in STE and the associated sub-categories of exchange between the reanalyses.

### 4.1    STE geographic distribution

Global spatial distributions of annually averaged STT and TST mass fluxes are shown in the right and left columns of Fig. 4, respectively. All of the models are similar regarding peak STT regions, with the largest differences found in contrasting the magnitudes of downward transport. Total STT mass fluxes are maximized along the Northern Hemisphere (NH) Atlantic

20    and Pacific extratropical storm tracks in each reanalysis model. Among the four reanalyses, ERA-Interim and JRA-55 STT mass fluxes are largest (Figs. 4a and 4c) with a maximum of $\sim$300 kg s$^{-1}$ km$^{-2}$ in the core of NH cyclone tracks. In the Southern Hemisphere (SH) the STT mass fluxes are largest within the subtropical latitudes along the western coasts of the continents. The largest SH STT maximum in each reanalysis is located along the subtropical coast of Chile. This is a confined but dominant region of STT mass flux, and is greatest in JRA-55 ($\sim$350 kg s$^{-1}$ km$^{-2}$) and weakest in MERRA ($\sim$200 kg s$^{-1}$

25    km$^{-2}$). Another common region of elevated STT in the SH occurs over a broad area in the extratropics along the west coast of Antarctica. A noteworthy difference among reanalyses regarding STT mass fluxes is found within the tropics, where the location of peak STT mass flux varies considerably. Peak regions of tropical STT are found along the equator across the Indian ocean in JRA-55 and ERA-Interim, but are found to be displaced south of the equator in both MERRA reanalyses.

Annually averaged geographic distributions of TST mass fluxes show considerably larger magnitudes and broader maxima

30    within the tropical latitudes compared to STT (right column Fig. 4). Spatially, maxima in tropical TST mass fluxes coincide with minima in STT mass flux. Distinct maxima of tropical TST mass fluxes are evident in the South China Sea, the West Pacific, the Caribbean Sea, and Southeast Asia. Within the subtropics a narrow band of TST extends from China into the east

Pacific and is consistent among the reanalyses. Comparison of TST among the reanalyses reveals two modes: regionally (or latitudinally) symmetric in ERA-Interim and JRA-55, and regionally asymmetric in MERRA and MERRA-2. The regional asymmetry in MERRA and MERRA-2 is due to significantly enhanced TST mass fluxes in the extratropics compared to the remaining reanalyses, with extratropical TST near $\sim$400 kg s$^{-1}$ km$^{-2}$ globally in MERRA and MERRA-2 and near $\sim$200 kg s$^{-1}$ km$^{-2}$ in ERA-Interim and JRA-55. Another inconsistency among the reanalyses is the location of peak TST in the tropical western Pacific. The peak tropical TST location in JRA-55 and ERA-Interim is highest in the NH but extends across the equator, whereas MERRA and MERRA-2 show a peak located mostly in the NH.

To better understand the differences and similarities of STT and TST geographic distributions, we can decompose both transport directions into separate regions of exchange to determine their contributions to the differences observed in total STE. The extratropical spatial distributions of STT and TST are shown in Fig. 5. One significant characteristic shown is the dominance of STT over TST in the extratropics indicated by ERA-Interim, JRA-55, and MERRA-2, especially over the NH and SH storm tracks. MERRA, however, shows the opposite behavior. As in the global STT distributions, the largest STT mass fluxes in the extratropics coincide with the NH cyclone tracks. In the SH, the STT mass fluxes are largest poleward of 60°S. Also, STT maxima across the subtropical SH, particularly the Chilean coast, are detected but magnitudes are weaker than those in total STT. More generally, STT mass fluxes in the extratropics are largest in JRA-55 and weakest in MERRA. TST in the extratropics, similar to STT, is found to be spatially consistent amongst the reanalyses, with the only apparent differences being the localized maxima near 60°N in MERRA and MERRA-2 that are not observed in ERA-Interim or JRA-55. In addition, TST in the extratropics in MERRA is generally larger than the remaining reanalyses.

STT and TST mass fluxes and their geographic distributions within the tropics are shown in Fig. 6. While both TST and STT mass fluxes are similar in magnitude between the reanalyses, there are slight offsets in the location and width of the identified maxima. For example, no two models agree on the precise location, zonal extent, or meridional extent of the TST maxima in the western Pacific. The offsets in STT and TST maxima outlined in the discussion of Figure 4 above are also evident in the maps of STT in the tropics.

Lastly, Figure 7 shows geographic distributions of STT and TST in the subtropics (i.e., that occuring between the tropics and extratropics across the tropopause break). These maps reveal that transport in the subtropics is 1) generally weaker than transport in the tropics and extratropics in each model, 2) is dominated by poleward transport through the tropopause break in each hemisphere (i.e., TST), and 3) is preferentially distributed over the ocean basins and extends poleward downstream of the ocean basins (in agreement with known evolution of Rossby wave breaking events, e.g., see Fig. 9 from Homeyer and Bowman, 2013). In reference to the total STE geographic distributions, it is evident that the influence of poleward and equatorward transport in the subtropics directly corresponds to the maxima in total STT and TST, which is expected due to its association with the tropopause break.

## 4.2 STE totals

In an effort to quantitatively summarize some of the apparent differences identified in the analysis of geographic distributions, globally integrated and annually averaged STE mass fluxes from each reanalysis are provided in Table 1. Total STT mass flux is similar among JRA-55, ERA-Interim, and MERRA-2, while the STT mass flux in MERRA is at least 25% lower. In the other direction, TST mass flux totals are significantly higher in MERRA (∼51%) and MERRA-2 (∼29%) compared to those from ERA-Interim and JRA-55.

Net STE is expected to be near zero, or balanced, over a long time period as a result of mass continuity. However, all reanalyses result in a net exchange that is either positive or negative (i.e. net TST or STT, respectively). Net mass fluxes in JRA-55 and ERA-Interim are both negative (STT-dominant), but the net flux amounts to only ∼4% of the total flux (TST + STT). However, net mass fluxes in MERRA and MERRA-2 are both positive and amount to about 33% and 12% of their total fluxes, respectively.

Table 1 also includes the integrated mass fluxes of the three transport subregions. Net tropical STE fluxes are positive and of similar magnitude in all four reanalyses, while extratropical STE fluxes are negative (STT-dominant) in JRA-55 and ERA-Interim, marginally negative in MERRA-2, and positive (TST-dominant) in MERRA. Similar to tropical STE, net subtropical fluxes are all postive (TST-dominant), with magnitudes in MERRA-2 and MERRA two to three times as large as those in ERA-Interim and JRA-55.

## 4.3 STE meridional distributions

Annually and zonally integrated latitudinal distributions of STE (TST, STT, and net) are shown in Fig. 8. Meridional distributions, similar to geographic distributions, demonstrate the latitudinal dependence of STE and offer a more quantitative comparison of the regional differences. Once again, we show both total STE and that separated by the regions of transport in the meridional distributions (with tropical and extratropical STE combined in this case).

Total STE meridional distributions (Fig. 8a) show similar latitudinal variations in TST and STT in each model, but differences in the magnitude of each transport direction lead to large differences in net STE, especially in the extratropics. The meridional distributions of STE in the extratropics and tropics combined (Fig. 8b) and in the subtropics (Fig. 8c) demonstrate that the majority of these differences can be attributed to extratropical and tropical STE, though TST plays an important role in the subtropics and is clearly much larger in the MERRA and MERRA-2 reanalyses (especially in the NH).

## 4.4 Annual cycles

In addition to understanding regional differences in STE, examining differences in the seasonality of transport can be important for determining the significance of STE on UTLS composition throughout the year. Annual cycles of normalized STT and TST in the extratropics are presented in Fig. 9 and separated by hemisphere, while normalized annual cycles for the tropics are

shown in Fig. 10. Mass fluxes of STT and TST are normalized using the maximum and minimum monthly means over the 15-year period:

$$N_i = \frac{\overline{month_i} - MIN(\overline{month})}{MAX(\overline{month}) - MIN(\overline{month})} \tag{2}$$

Where, $\overline{month_i}$ is the monthly mean for each month ($i = 1$–12) and $N_i$ is now the $i^{th}$ normalized monthly mean mass flux.

Normalized STT mass fluxes are similar among the models in the NH and SH extratropics, but the hemispheres differ in the timing of annual minimum and maximum STT. Annual STT is at maximum and minimum in late winter (DJF) and late summer (JJA) in the NH, while the maximum and minimum STT occur during early autumn (MAM) and spring (SON) in the SH. One unique difference in STT is apparent in the SH during Austral summer, where MERRA-2 normalized mass fluxes are considerably smaller during those months compared to the other reanalyses. Although the reanalyses show similar

seasonality for STT, there are some differences in monthly variability. While there are no uniform differences in variability within the hemispheres or among the reanalyses, the monthly variabilities are considerably larger in the SH. Annual cycles of extratropical TST (bottom row of Fig. 9) have similar modes and hemispheric differences in monthly variability to that of STT, especially in the NH. However, there is an apparent 1 to 2 month offset in the minima and maxima of TST and STT in each hemisphere.

There are more apparent differences in the normalized annual cycles of TST and STT in the tropics (Fig. 10). In particular, normalized TST in the tropics reveals two preferred seasonal cycles. In JRA-55 and ERA-Interim TST is weakly bimodal with local maxima occurring during the NH winter (DJF) and summer (JJA). MERRA and MERRA-2, on the other hand, are broadly unimodal and reach a maximum during the NH summer and minimum in NH fall (SON). STT seasonality in the tropics is more consistent amongst the reanalyses, with general agreement of STT peaking in the late NH summer and early fall.

While the normalized annual cycles of TST and STT demonstrate the seasonality of STE, they do not represent amplitudes (i.e. $MAX(\overline{month}) - MIN(\overline{month})$) of seasonality among the reanalyses. Table 2 provides annual cycle amplitudes for each transport direction, transport region, and hemisphere. Annual cycle amplitudes are similar among the reanalyses for extratropical STT, subtropical TST, and tropical TST, while largely different for extratropical TST (especially in the NH), subtropical STT, and tropical STT.

Annual cycles of net STE are shown in the top row of Figure 11 and left un-normalized to show the combined effects of differences in seasonality and in dominance of STE pathway. MERRA and MERRA-2 show a positive net cross-tropopause mass flux (TST) throughout their annual cycles in the NH. Alternatively, JRA-55 and ERA-Interim exhibit a NH seasonal cycle that is STT-dominant in the winter and early spring and TST dominant in the summer. In the SH, JRA-55 and ERA-Interim again show consistent seasonality, and are STT-dominant through most of the year and only briefly positive during the

summertime (DJF). While similar in shape, MERRA-2 exhibits positive net exchange that spans all seasons but winter (JJA). MERRA, as in the NH, exhibits only positive net STE and a weaker annual cycle compared to the other reanalyses. The largest monthly net STE mass flux variability is found in the annual cycles of MERRA.

Annual cycles of the transport categorized by region also reveal important differences in the seasonality of STE and the contribution of individual processes to the total annual cycles. Net STE annual cycles in the extratropics and tropics (combined)

and in the subtropics are given in Fig. 11 for both hemispheres. Generally, the seasonality of net mass flux in the extratropics and tropics is similar in shape to the net annual cycle of total STE in each hemisphere, with a few important differences. Specifically, the combined extratropics and tropics net STE annual cycle of MERRA-2 is shown to be STT-dominant during the NH wintertime, whereas the total net STE indicates positive net exchange through all seasons. The SH combined extratropics and tropics net STE annual cycles are not significantly different from those represented by total net STE, aside from a negative shift (i.e., a greater influence of STT). Figure 12 shows tropical exchanges removed from the extratropics and reveals that annual cycles of STE in the tropics are similar among the reanalyses. This demonstrates that the separation in STE annual cycles in the combined hemispheric analysis (middle row of Fig. 11) is due to STE in the extratropics.

For annual cycles of transport in the subtropics, all four reanalyses show similar seasonal behavior, with a minimum in late spring and early summer and a maximum during the late fall and early winter and are TST dominant in both hemispheres. Annual cycles of subtropical transport in MERRA and MERRA-2 have a slightly larger amplitude than those in ERA-Interim and JRA-55 and are displaced at higher net positive fluxes, which is consistent with the analyses presented in Sections 4.1 & 4.3 above.

## 4.5 STE time series

Time series of STE are examined over the 15-year period to further evaluate similarities and differences between the reanalysis models. In Fig. 13, global time series of STT anomalies and TST anomalies are shown with respect to their mean mass fluxes (i.e., 15-yr means are removed). STT throughout the period shows two modes of long-term changes in the reanalyses: increasing mass fluxes over time in JRA-55 and ERA-Interim and decreasing mass fluxes over time in MERRA and MERRA-2. Similar long-term increases in TST can be seen in the ERA-Interim and JRA-55 time series. In the MERRA reanalyses, however, long-term changes in TST are shown to be largely increasing in MERRA and near-zero in MERRA-2.

STE time series separated by region (combined extratropical and tropical, and subtropical) are shown in the middle and bottom rows of Fig. 13. Combined extratropical and tropical STT shows similar patterns and variability to that from total STT for each reanalysis. Mainly, there are decreasing combined extratropical and tropical STT mass fluxes in MERRA and MERRA-2 and increasing fluxes in JRA-55 and ERA-Interim. In a similar manner, combined extratropical and tropical TST is increasing over the period in ERA-Interim, JRA-55, slightly decreasing in MERRA-2, and strongly increasing in MERRA. Subtropical STE time series for all reanalyses show little to no long-term variability. There is one exception, however, with subtropical TST in MERRA showing a potential long-term increase in mass flux.

The long-term increases and decreases in STE identified in Figure 13 are associated primarily with extratropical and tropical STE and are considerably large relative to the mean, especially given the relatively short time period analyzed. In order to better understand the source of these changes we also analyzed time series of extratropical and tropical STE separately (Fig. 14). Based on the analyses presented thus far and conventional STE knowledge, transport in the tropics is primarily upward (TST), while it is primarily downward (STT) in the extratropics (with the exception of MERRA). These geographically separated upward STE modes are largely the result of the BDC. Thus, we expect long-term changes in TST in the tropics and STT in

the extratropics to be consistent. Figure 14 demonstrates this well, with negligible long-term changes in STT in the tropics and apparent STT changes in the extratropics, and the opposite behavior for TST (except for MERRA). Specifically, JRA-55 and ERA-Interim show increasing STT in the extratropics and TST in tropics, whereas MERRA-2 shows decreasing mass fluxes over the 15-year period.

The observed consistency in the sense of the long-term changes of TST in the tropics and STT in the extratropics in ERA-Interim, JRA-55, and MERRA-2 suggests that changes in the BDC may be responsible for this behavior. Specifically, the increasing fluxes for tropical TST and extratropical STT over time in JRA-55 and ERA-Interim indicate an acceleration in the BDC, while the decreases in MERRA-2 indicate a deceleration of the BDC. Changes in the speed of the BDC have been examined in previous studies. For example, Abalos et al. (2015) evaluated the dynamics of the BDC using ERA-Interim, JRA-55,

and MERRA and show that there is general agreement in a strengthening BDC over the period 1979-2012 by 2-5% per decade. Observational studies show decreases in tropical stratospheric water vapor, ozone, and temperature observed by satellite, which also corresponds to an increase in tropical upwelling associated with an accelerated BDC (Randel et al., 2006). Chemistry-climate models have also indicated an acceleration of the BDC over time (e.g., Austin and Li, 2006). These previous reanalysis, observational, and modeling studies are consistent with the results from ERA-Interim and JRA-55 here, while MERRA-2 is in

disagreement and MERRA does not indicate changes in the BDC over time in our analysis.

## 4.6   Reanalysis model evaluations

This paper is largely a comparison of STE estimates using multiple state-of-the-art reanalysis models, but we also briefly evaluate some model differences through various diagnostics here. The goal is to provide general context and logical reasoning to

20 explain some of the aforementioned STE differences among the reanalyses.

### 4.6.1   STE occurrence geographic distributions

In order to assess quantitative and qualitative STE differences, particularly the larger differences found among the models' representation of TST, we have to consider whether the frequency of STE events differs among the models. Evaluating STE

occurrence frequency in each reanalysis informs us whether the amount of STE is a result of more frequent STE or differences in the magnitude of transport in individual events.

    In Fig. 15, total STT and TST occurrence frequencies are shown. There are noticeable differences between the ERA-Interim and JRA-55 pair and the MERRA reanalyses. In particular, ERA-Interim and JRA-55 show higher occurrence frequencies globally for STT, while the MERRA and MERRA-2 occurrence frequencies are higher for TST. Taken together with the re-

30 sults from the geographic distributions of STE mass flux (Figs. 4–7), these analyses suggest that differences in mass flux between the reanalyses are largely the result of differences in the frequency of exchange events.

### 4.6.2 Other Diagnostics

The differences in STE occurrence and mass flux estimates among the models, to some extent, are due to dynamical and/or physical differences between the models. Over a long period these small differences may result in considerable variations in climatological evaluations of STE. Dynamical differences may include variability in vertical winds among the reanalyses and in the strength of the subtropical and polar jets. Physical differences, for example, include the altitude of the tropopause and its variability among the reanalyses. These dynamical and physical characteristics can impact a reanalysis model's short- and long-term representation of STE. For example, differences in STE can be the result of higher or lower tropopause altitudes among the models if the position and strength of the 3-D wind fields are similar in the UTLS.

As the largest differences in our comparison are those associated with STE in the extratropics, comparing the magnitudes of vertical motion at the tropopause may reveal a dynamical source of transport differences. In Fig. 16, probability density functions (PDFs) of vertical motion are shown for each season and separated into the NH extratropics, SH extratropics, and tropics (for a single year - 2003, but note that additional years are similar). While these PDFs demonstrate well that vertical motion is a bit stronger (i.e., more frequent extremes) in JRA-55 and ERA-Interim, there aren't necessarily clear differences in the skewness of these PDFs that support the differences in net STE outlined previously. This result may suggest that differences in quasi-isentropic exchange are a source for the observed differences. It is important to note, however, that the strongest vertical winds (i.e. extremes of the PDFs) have higher frequencies in the SH extratropics, which may help to explain the larger STE variability observed there.

Physical differences among the reanalyses are very important for STE studies, since identifying STE in the first place requires a definition of the troposphere-stratosphere boundary (i.e., LRT or PV isosurface). To examine such differences, we evaluate global PDFs of tropopause pressure from each reanalysis for the entire 15-yr period analyzed in this study, separated by season (Fig. 17). These PDFs show a bimodal distribution in each season, with a tropical mode at pressures less than 150 hPa and an extratropical mode at pressures greater than 150 hPa (as outlined previously). However, there are consistent differences between the models that are clear in the extratropical mode of the distribution. In particular, extratropical tropopause pressures are skewed to lower values (or higher altitudes) in MERRA and MERRA-2 compared to those in JRA-55 and ERA-Interim. These differences suggest that offsets in the altitude of the tropopause may be an important contributor to the dichotomy in net STE observed between these two pairs of reanalyses, which was found to be greatest in the extratropics.

## 5    Conclusions and discussion

In this study, we examined global characteristics of STE over a 15-year period (1996-2010) using a trajectory model and output from multiple reanalyses: ERA-Interim, JRA-55, MERRA-2, and MERRA. STE was separated into three regions based upon the altitude of the tropopause in an attempt to isolate known transport processes associated with STT and TST.

## 5.1 Principal conclusions

In contrast to the vast majority of previous work, this study used the lapse-rate tropopause or LRT as the troposphere-stratosphere boundary rather than an isosurface of potential vorticity (PV) or dynamic tropopause. In order to demonstrate the impact of this choice for STE studies, we presented a comparison of STE estimates using the LRT method and results from a recent study that used a dynamic tropopause (Škerlak et al., 2014). We found that:

1. magnitudes of STE are uniformly smaller using the LRT,

2. spatial placement and variability of STT is similar between methods, and

3. spatial placement and variability of TST is largely different, with the most significant differences found in the polar regions.

These differences correspond to a change in the net transport direction in the polar regions when using the LRT (i.e., STT-dominant rather than TST-dominant, though the magnitudes in each case are small compared to the global amounts). Such net transport at high latitudes from the LRT method is more consistent with our established understanding of UTLS dynamics: net TST in the tropics and net STT in the extratropics and polar regions.

The main focus of this paper was not a method comparison, but a comparison of STE among four state-of-the-art atmospheric reanalyses. Doing so, we separated transport into several regions in order to investigate the STE climatologies both quantitatively and qualitatively. It was found that the models can be grouped into two populations: STT-dominant and TST-dominant (Table 1). JRA-55 and ERA-Interim are STT-dominant, while MERRA and MERRA-2 are TST-dominant. The net transport in the STT-dominant reanalyses, however, is small relative to the total transport, while the opposite is true for the TST-dominant reanalyses.

Geographic distributions and zonal mean latitudinal distributions revealed important characteristics about the two reanalysis populations. Notably, the largest differences in STE were found in the extratropics. Geographic distributions of STT maxima were similar amongst all reanalyses, while the opposite was true for TST. MERRA was typically an outlier relative to the remaining reanalyses, but similar differences (though largely diminished) were found between MERRA-2 and the STT-dominant reanalyses (ERA-Interim and JRA-55). STE in the subtropics was found to be consistent geographically with prior studies of Rossby wave breaking events along the tropopause break (e.g., Postel and Hitchman, 1999; Homeyer and Bowman, 2013). Although geographic placement and net transport for STE in the subtropics was consistent among the models, the MERRA reanalyses showed roughly twice the magnitude of net poleward transport from the tropical troposphere into the extratropical lowermost stratosphere (i.e., TST).

Seasonality of STE amongst the reanalyses was also found to be similar for some transport regions and directions and significantly different for others. Similar to geographic consistencies observed, we found that STT and TST annual cycles in the extratropics are consistent among the reanalyses and in both hemispheres. However, seasonality of TST in the tropics was found to be weakly bimodal in STT-dominant reanalyses and unimodal in the TST-dominant reanalyses, while STT seasonality in the tropics showed smaller differences. Larger differences were found for annual cycles of net STE from the reanalyses, with

MERRA showing little seasonality in each hemisphere. Differences in net STE were shown to be associated primarily with extratropical STE in each reanalysis.

Long-term changes were also investigated using time series analysis over the 15-year study period. These analyses indicated gradual increases and decreases in STT and TST mass flux for the STT-dominant models and MERRA-2, respectively. Further analyses suggested that long-term changes in total STE are associated with either an acceleration or deceleration of the BDC. Specifically, the BDC is apparently decelerating in MERRA-2 and accelerating in JRA-55 and ERA-Interim from 1996–2010.

Finally, several diagnostics were applied to the reanalyses in order to shed light on the sources of the STE differences. We found that differences in transport are likely the result of differences in the frequency of irreversible STE rather than the magnitude of individual events. We also found the altitude of the tropopause between the STT-dominant and TST-dominant models to differ considerably in the extratropics. An analysis of vertical winds at the tropopause showed some differences between the TST-dominant and STT-dominant reanalyses, but these were not necessarily consistent with the nature of the STE differences found. Taken together, these physical and dynamical differences may be significant sources of variability for climatological analyses and it is likely that they contribute to some of the STE differences observed in this study.

## 5.2 Discussion

The analyses in this study demonstrate that while there are some areas of agreement in the magnitude, geographic distribution, and frequency of large-scale STE among modern reanalyses, there are important differences that can lead to varying conclusions of the impact of STE on UTLS composition, the radiation budget, and climate. While this study is the first model comparison of global STE estimates, there are some limitations that could be improved upon in future work. First, the analysis time period could be increased. Each reanalysis model used in this study has output available from 1979 to 2015, roughly 2.5 times longer than that used here. Expanding the analysis period may provide improved confidence in the statistical behavior of STE regarding the long-term changes associated with the BDC (Section 4.5). An extended analysis period may also reduce variability in the seasonality and regional distributions analyzed here and thus increase confidence in those results. The primary challenges with this suggested expansion in the analysis time period are the computational time required and the cost of data storage (15 years of 6-hour model output and trajectory model calculations from the four reanalyses used here requires $\sim$10 TB of disk storage).

Second, while the present generation of the reanalysis models are significant advancements for studies of UTLS dynamics and associated processes over previous generations, model improvements can still be made in the UTLS. Given the limited spatiotemporal observations of STE available, it is understandable that model simulations of transport could differ considerably. However, some of these differences are likely related to basic model choices such as grid resolution. For example, the vertical grids are nearly equivalent in ERA-Interim and JRA-55, which differ considerably from that used in MERRA and MERRA-2. Notably, vertical resolution is finer at levels below the tropical tropopause in ERA-Interim and JRA-55, but finer above the tropical tropopause in the MERRA Reanalyses. Since the vertical grid resolution and placement of vertical grid levels in the UTLS may have important impacts on tropopause-relative analyses such as STE (especially since current vertical resolution of reanalyses in the UTLS is $\sim$1 km), it is important to understand the sensitivity of such analyses to this model design choice.

Third, as referred to in the Introduction, large quantitative uncertainties in STE exist from previous Eulerian and Lagrangian STE studies. In particular, estimates for STE have often been limited to specific regions or time periods or based on inadequate and/or incomplete methods (compared to that possible with current methods and computational abilities). Here, we found that mean net STE magnitudes also range considerably when an equivalent method is applied to multiple modern reanalyses (e.g.,

see Table 1). However, few previous studies enable direct comparison with our estimates. In particular, the alternative PV-based approach by Škerlak et al. (2014) is arguably the most direct, where ERA-Interim net STE integrated globally over the 15 yr period in our study is approximately $1.48 \times 10^{17}$ kg/yr downward (STT), while it is about 3.5 times smaller ($4.2 \times 10^{16}$ kg/yr) in the Škerlak et al. (2014) study. These differences do not necessarily suggest that one method is superior to the other, but that two largely similar Lagrangian approaches can yield substantially different results due to the employed troposphere-stratosphere

boundary (e.g., see Figs. 2 & 3).

Lastly, while this study compared STE among reanalyses and attempted to diagnose potential sources of those differences, much more work can be done to examine them. The clearest source of STE differences found was related to STT and TST event frequencies (Fig. 15), but differences in the vertical winds and extratropical LRT distributions shown in Figs. 16 & 17 may imply that the dynamical regimes evaluated for exchange in each reanalysis are slightly different. One approach to further

evaluate the role of these differences in dynamics and boundary conditions may be to combine tropopause altitudes from one reanalysis with the winds of another reanalysis for STE trajectory calculations. Such analyses may lead to future improvements in the model grids and numerics, especially in the UTLS.

*Acknowledgements.* We thank the agencies that provided meteorological data used in this study: JRA-55 from JMA and ERA-Interim from

20 ECMWF both obtained from Reanalysis Data Archive (RDA) managed by Computational and Information Systems Laboratory (CISL) at the National Center for Atmospheric Research (NCAR), and MERRA and MERRA-2 from NASA. We also thank Bojan Škerlak and Michael Sprenger for providing output from their 2014 paper to be used for comparison with our work.

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

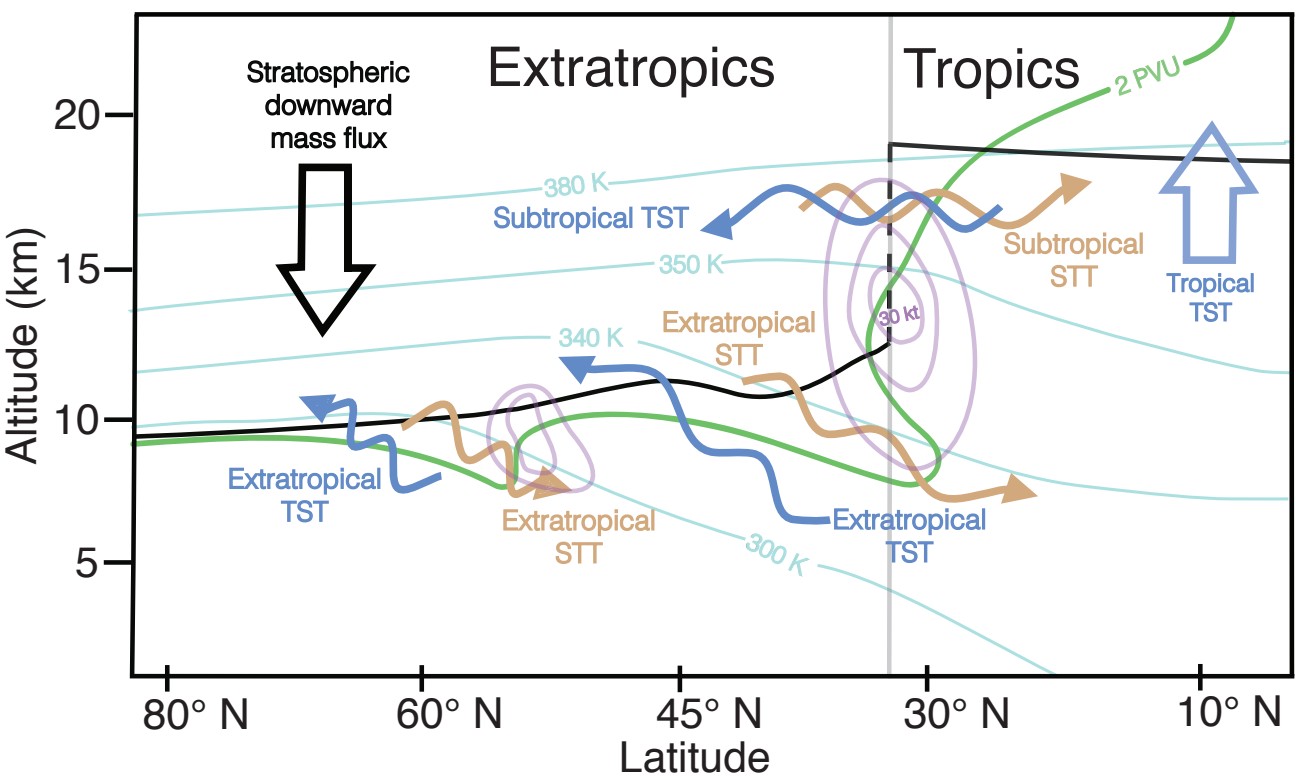

**Figure 1.** Hemispheric schematic of the STE identification method employed in this study. STT and TST are shown by orange and blue arrows, respectively. Bold and wavy arrows represent the Brewer–Dobson circulation and quasi-isentropic mixing processes, respectively. Potential temperature surfaces are given by the cyan lines, the lapse-rate tropopause by the black lines, and the 2 PVU potential vorticity (PV) isosurface by the green line. The solid gray line, coincident with the tropopause break (dashed black line), represents the boundary between the tropics and extratropics. Subtropical and polar jet streams are shown by solid purple isotachs.

# STT

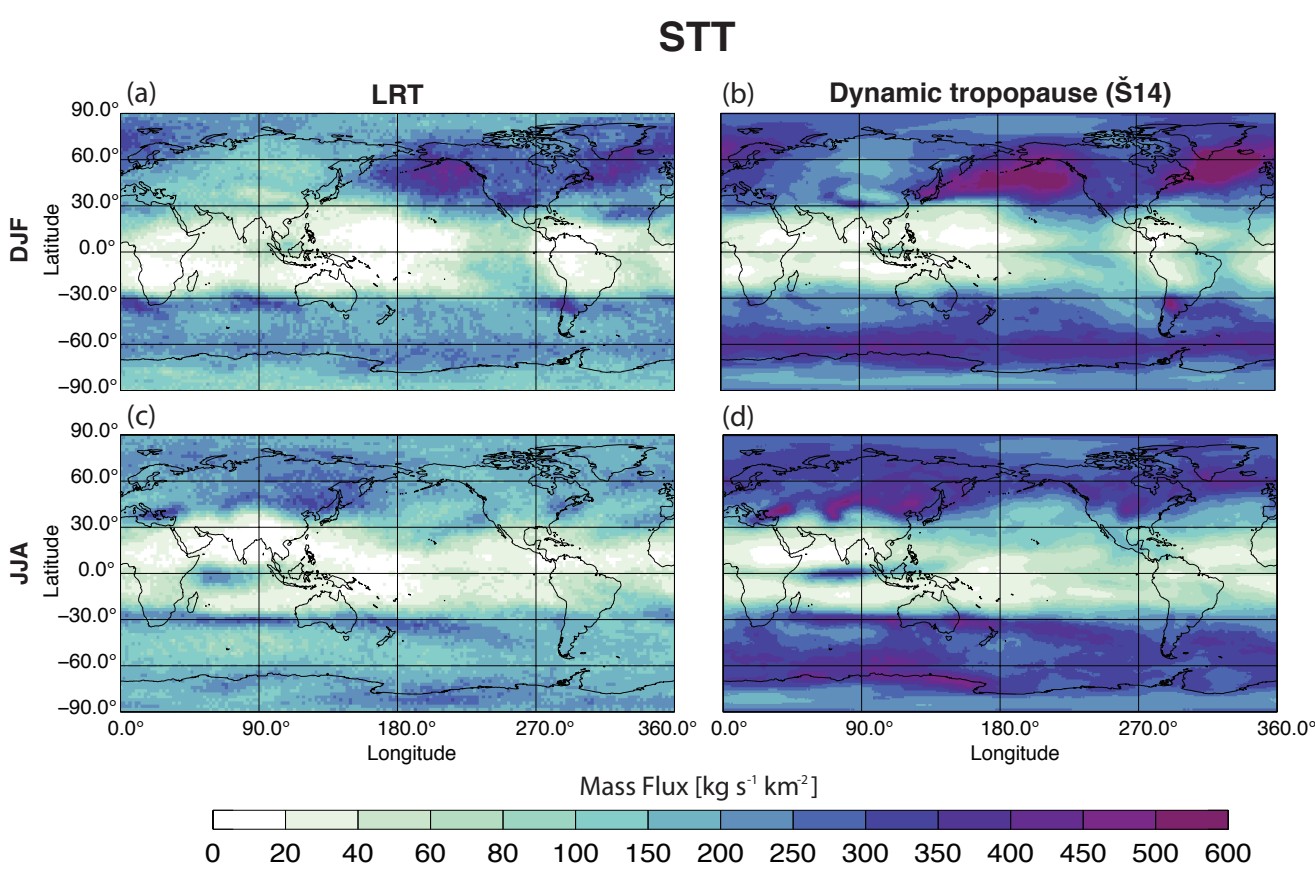

**Figure 2.** December, January, and February (DJF, top row) and June, July, and August (JJA, bottom row) mean STT mass fluxes for 1996–2010 using the lapse-rate tropopause method (left) and 1979–2011 using the dynamic tropopause method (right; from Škerlak et al. (2014))

.

# TST

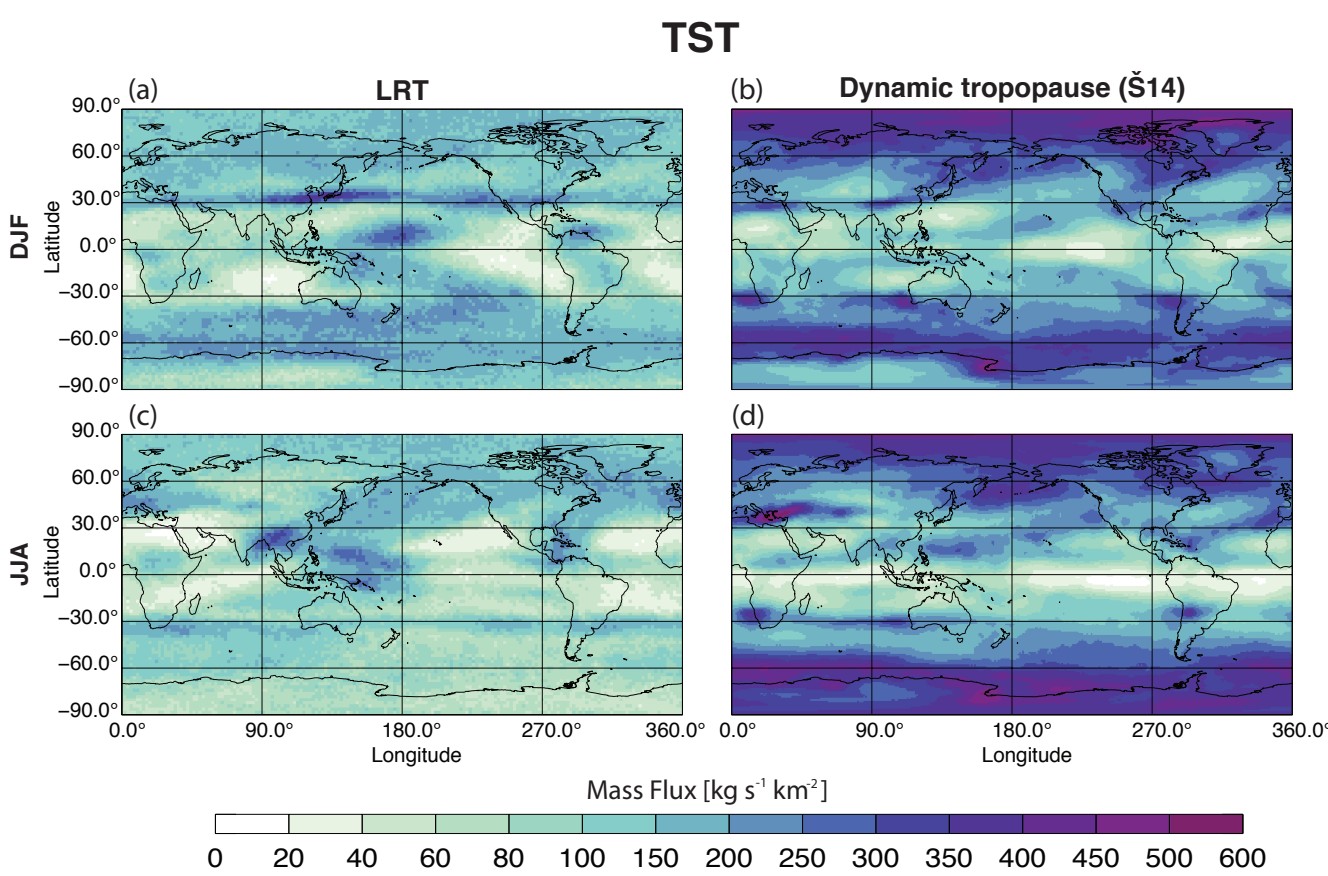

**Figure 3.** As in Fig. 2, but for TST.

# Total STT

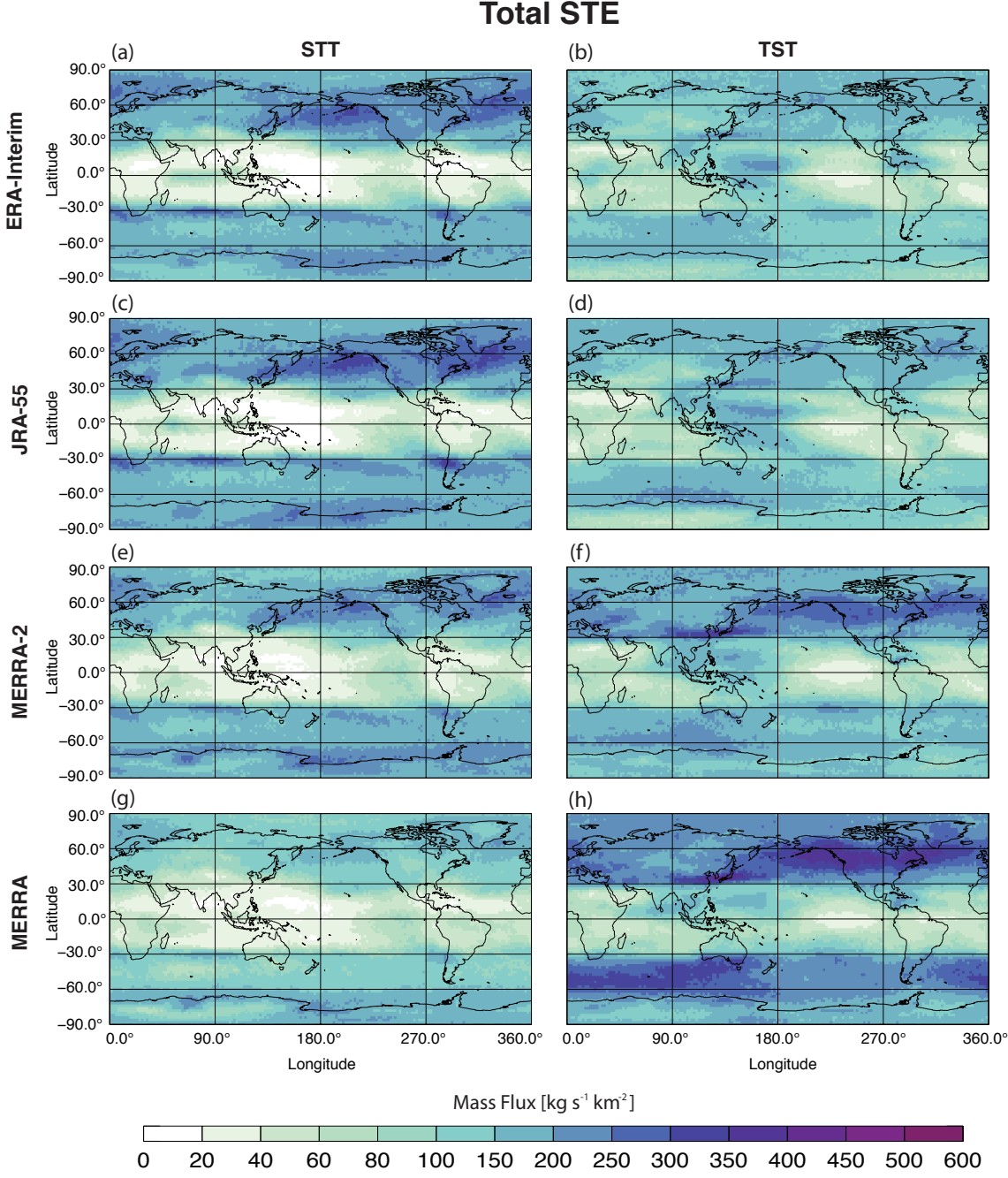

**Figure 4.** Annual mean geographic distributions of total (sum of tropical, subtropical, extratropical) STE mass flux for ERA-Interim (a & b), JRA-55 (c & d), MERRA-2 (e & f), and MERRA (g & h). STE is separated into STT (left) and TST (right) for each reanalysis.

# Extratropical STE

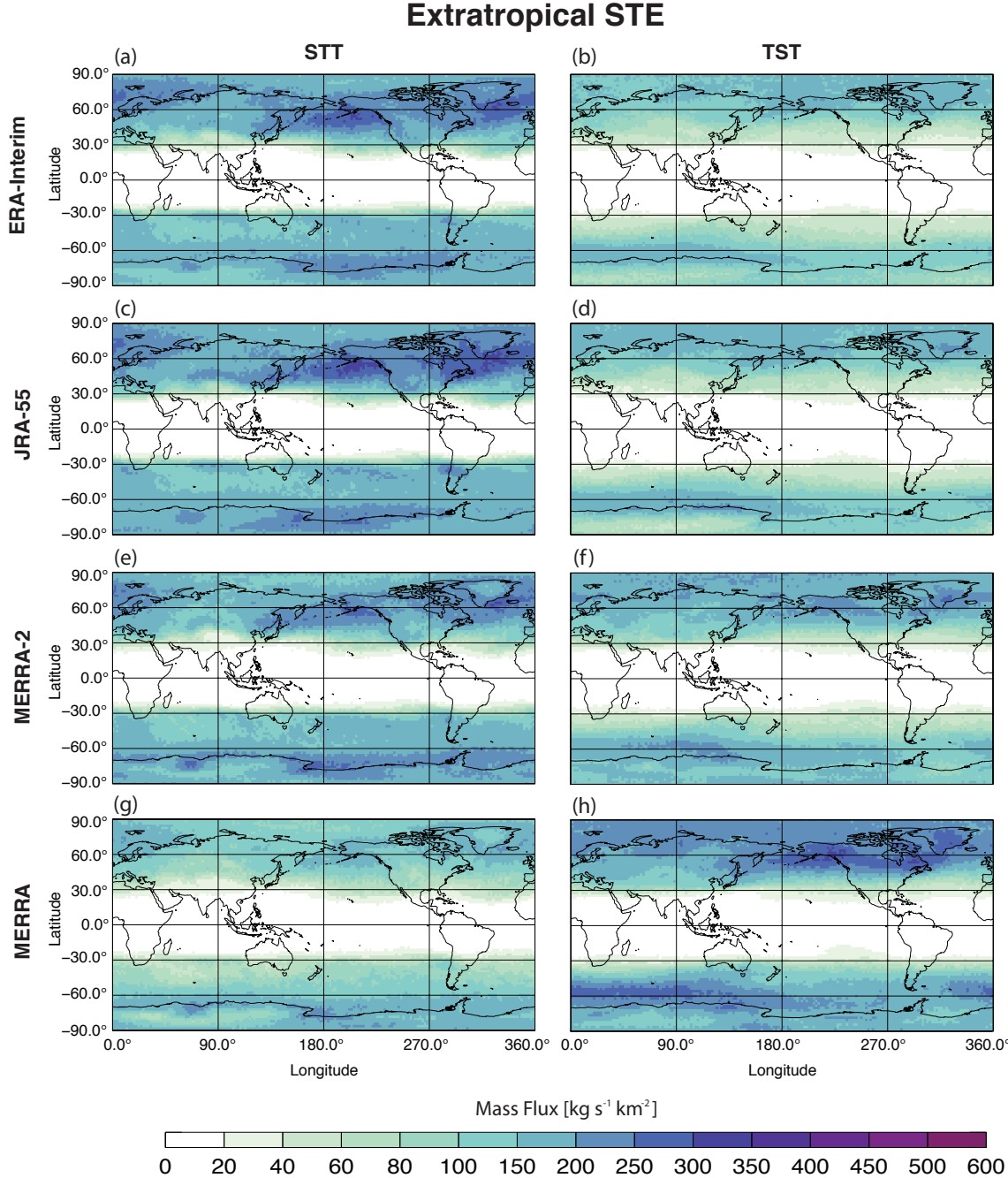

**Figure 5.** As in Fig. 4, but for STE in the extratropics.

# Tropical STE

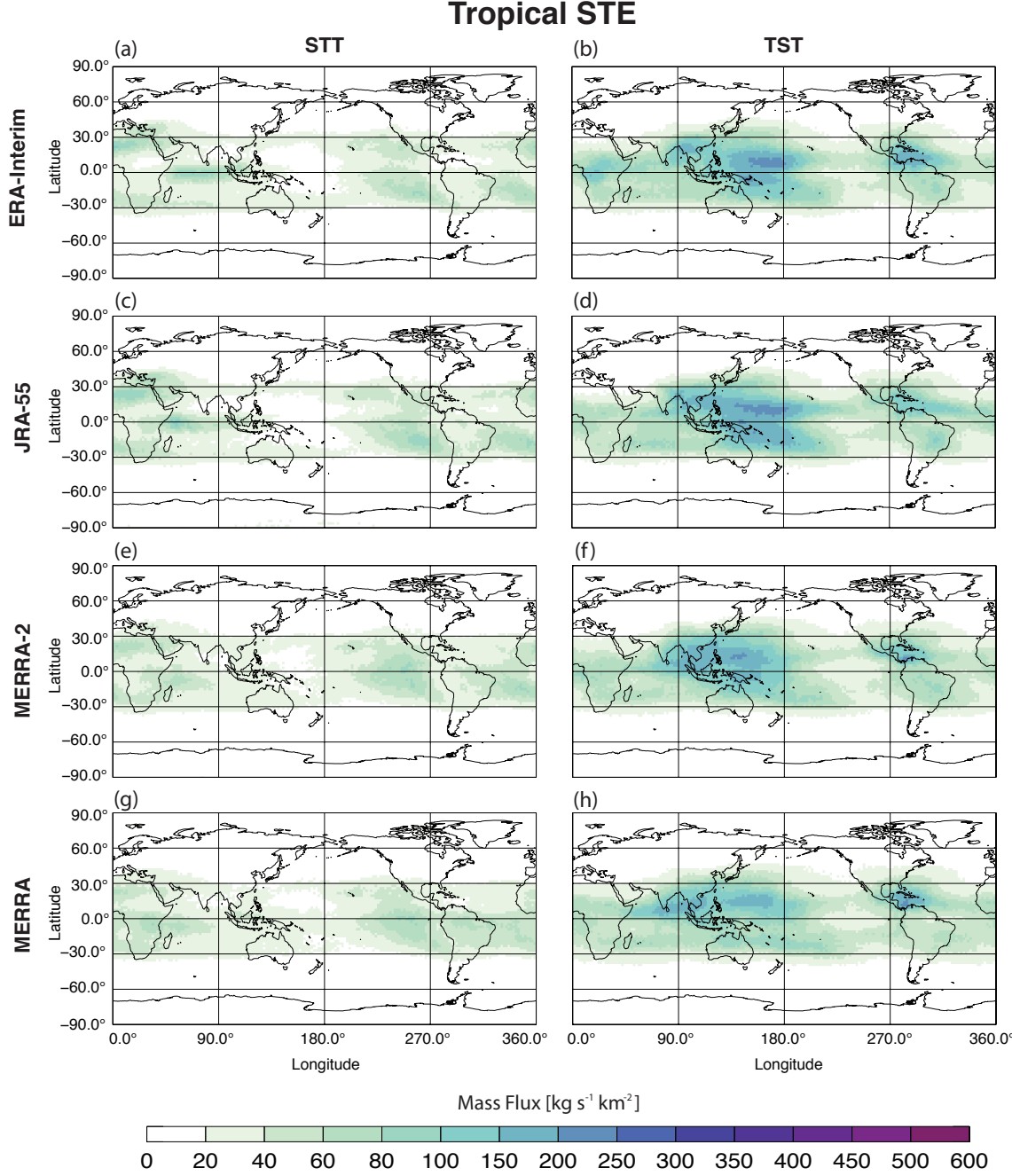

**Figure 6.** As in Fig. 4, but for the STE in the tropics.

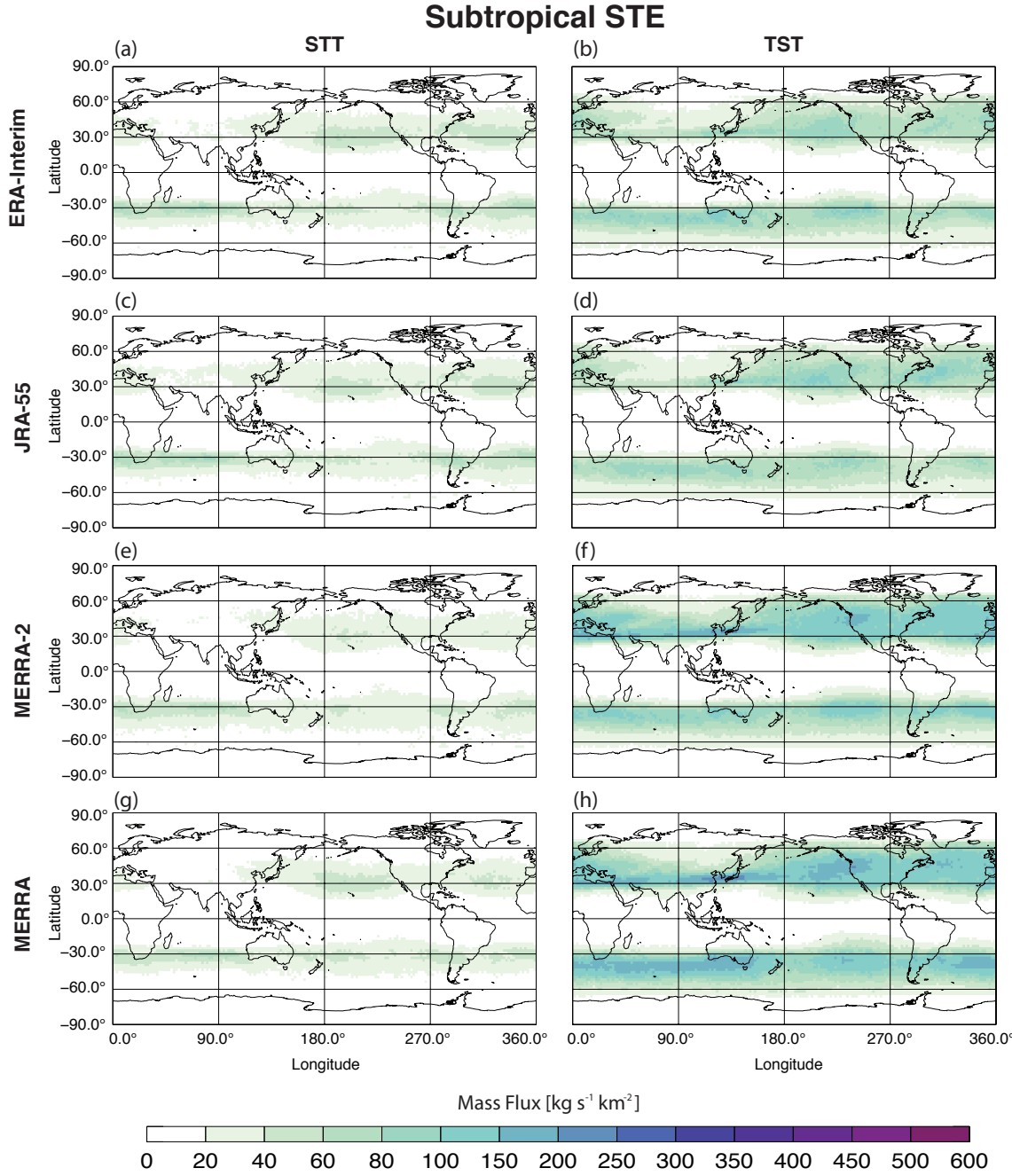

**Figure 7.** As in Fig. 4, but for STE in the subtropics.

**Table 1.** Globally integrated and annually averaged STE mass fluxes for each reanalysis model. STT, TST, net (TST-STT), and gross (TST+STT) mass fluxes are given for each transport region (i.e. total/global, extratropical, tropical, and subtropical). All mass flux units are $10^{10}$ kg s$^{-1}$.

| Total (Global) STE | | | | |
| --- | --- | --- | --- | --- |
| Reanalyses | STT | TST | Net | Gross |
| JRA-55 | 5.83 | 5.29 | -0.54 | 11.12 |
| ERA-Interim | 5.56 | 5.10 | -0.47 | 10.66 |
| MERRA-2 | 5.16 | 6.57 | 1.41 | 11.73 |
| MERRA | 3.86 | 7.74 | 3.88 | 11.60 |

| Extratropical STE | | | | |
| --- | --- | --- | --- | --- |
| Reanalyses | STT | TST | Net | Gross |
| JRA-55 | 4.53 | 2.30 | -2.23 | 6.82 |
| ERA-Interim | 4.29 | 2.17 | -2.12 | 6.47 |
| MERRA-2 | 3.94 | 2.97 | -0.97 | 6.92 |
| MERRA | 2.65 | 3.91 | 1.26 | 6.57 |

| Tropical STE | | | | |
| --- | --- | --- | --- | --- |
| Reanalyses | STT | TST | Net | Gross |
| JRA-55 | 0.77 | 1.81 | 1.04 | 2.58 |
| ERA-Interim | 0.72 | 1.69 | 0.96 | 2.41 |
| MERRA-2 | 0.75 | 1.71 | 0.96 | 2.45 |
| MERRA | 0.74 | 1.64 | 0.90 | 2.38 |

| Subtropical STE | | | | |
| --- | --- | --- | --- | --- |
| Reanalyses | STT | TST | Net | Gross |
| JRA-55 | 0.54 | 1.19 | 0.65 | 1.73 |
| ERA-Interim | 0.55 | 1.24 | 0.69 | 1.79 |
| MERRA-2 | 0.47 | 1.89 | 1.42 | 2.36 |
| MERRA | 0.47 | 2.19 | 1.72 | 2.66 |

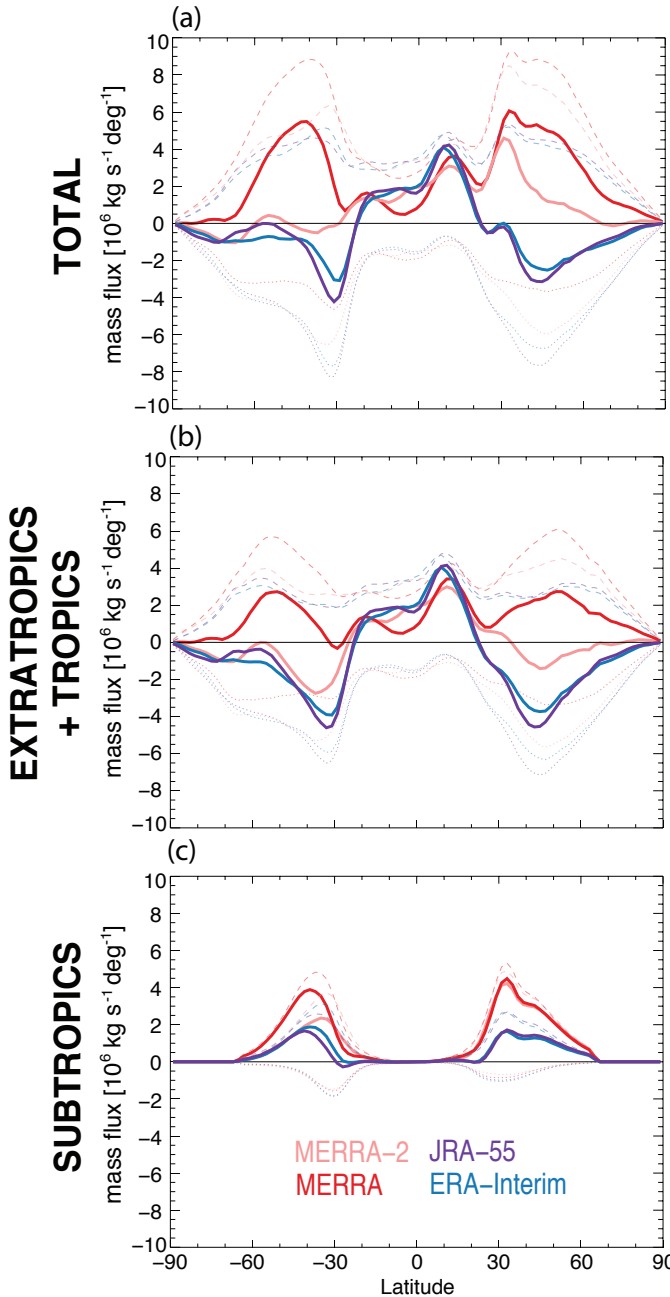

**Figure 8.** Annually and zonally averaged meridional distributions of STE from each reanalysis as a function of latitude and for (a) total STE, (b) extratropical and tropical STE combined, and (c) subtropical STE. STT is shown as the dotted lines (negative), TST as the dashed lines (positive), and the net transport is given by the solid lines in each panel. STE from JRA-55 is shown by the purple lines, ERA-Interim by the blue lines, MERRA-2 by the light red lines, and MERRA by the dark red lines.

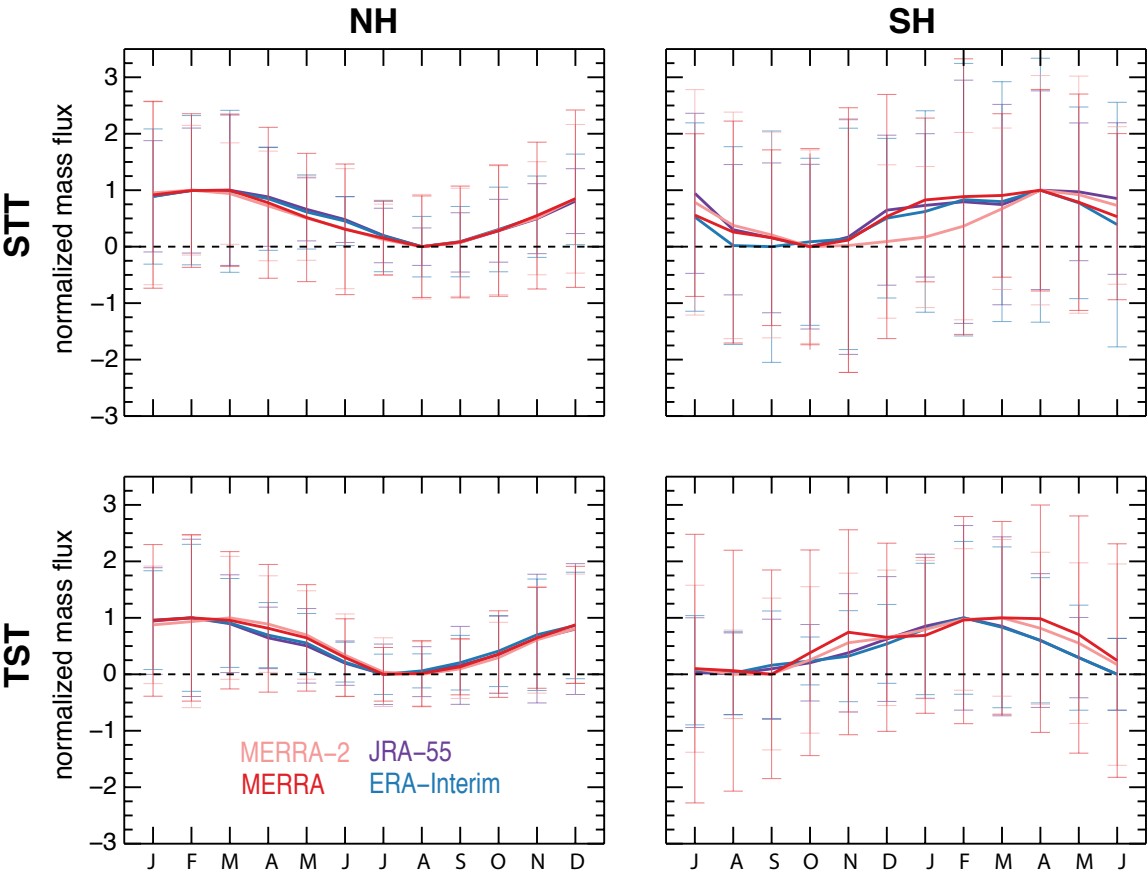

**Figure 9.** Annual cycles of (top row) normalized STT and (bottom row) normalized TST for the (left) extratropical Northern Hemisphere and (right) extratropical Southern Hemisphere from each reanalysis model. In each plot, the solid colored lines are the mean annual cycles and the colored error bars are plus/minus one standard deviation from the mean. STE from JRA-55 is shown by the purple lines, ERA-Interim by the blue lines, MERRA-2 by the light red lines, and MERRA by the dark red lines. Note that NH and SH annual cycles are offset by 6 mo.

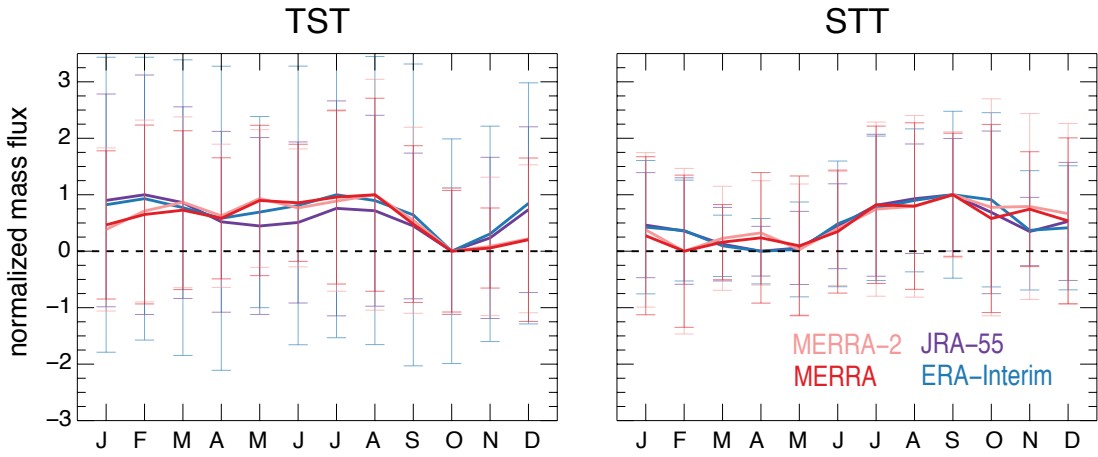

**Figure 10.** Annual cycles of (left) normalized TST and (right) normalized STT for the tropics from each reanalysis model. As in Figure 9, the solid colored lines are the mean annual cycles and the colored error bars are plus/minus one standard deviation from the mean. STE from JRA-55 is shown by the purple lines, ERA-Interim by the blue lines, MERRA-2 by the light red lines, and MERRA by the dark red lines.

**Table 2.** STT and TST annual cycle amplitudes in the extratropics, subtropics, and the tropics from each reanalysis model. All amplitudes are in units of $10^9$ kg s$^{-1}$ and annual cycles for the extratropics and subtropics are separated by hemisphere.

| Reanalysis | $\mathbf{STT}_{\text{Ex}}$ | | $\mathbf{TST}_{\text{Ex}}$ | |
|---|---|---|---|---|
| | NH | SH | NH | SH |
| JRA-55 | 13.18 | 4.91 | 6.13 | 12.92 |
| ERA-Interim | 14.19 | 3.90 | 6.23 | 10.68 |
| MERRA-2 | 14.88 | 5.94 | 11.4 | 12.92 |
| MERRA | 10.64 | 6.37 | 13.61 | 10.47 |

| Reanalysis | $\mathbf{STT}_{\text{Sub}}$ | | $\mathbf{TST}_{\text{Sub}}$ | |
|---|---|---|---|---|
| | NH | SH | NH | SH |
| JRA-55 | 1.11 | 3.49 | 2.70 | 6.07 |
| ERA-Interim | 1.36 | 3.76 | 2.89 | 5.71 |
| MERRA-2 | 0.67 | 7.01 | 2.85 | 4.21 |
| MERRA | 0.96 | 8.19 | 2.88 | 5.50 |

| Reanalysis | $\mathbf{STT}_{\text{Tropic}}$ | $\mathbf{TST}_{\text{Tropic}}$ |
|---|---|---|
| JRA-55 | 5.05 | 5.92 |
| ERA-Interim | 4.67 | 5.91 |
| MERRA-2 | 2.06 | 4.23 |
| MERRA | 2.35 | 5.03 |

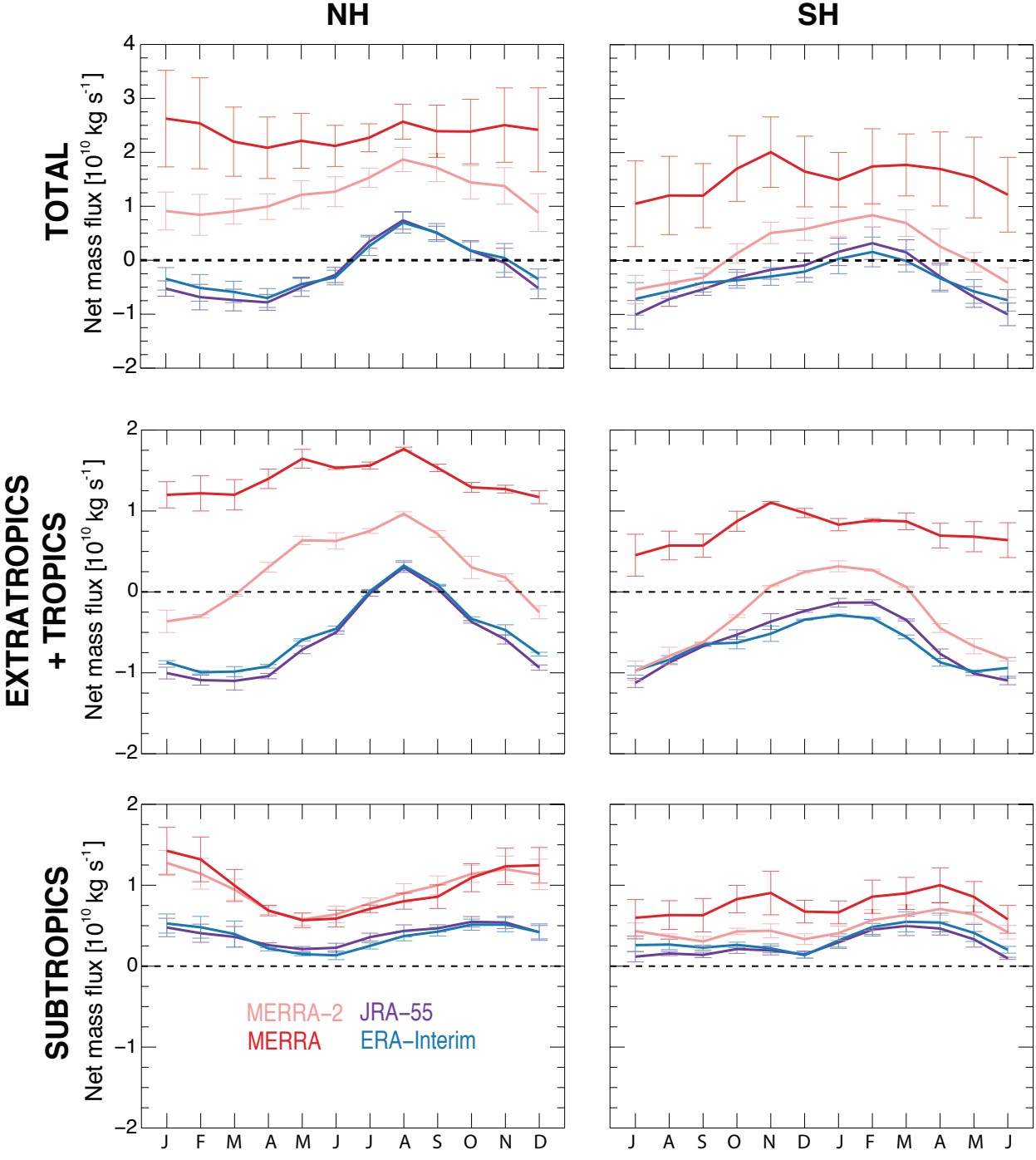

**Figure 11.** As in Figure 9, but for non-normalized total net STE (top row), combined extratropical and tropical net STE (middle row), and subtropical net STE (bottom row).

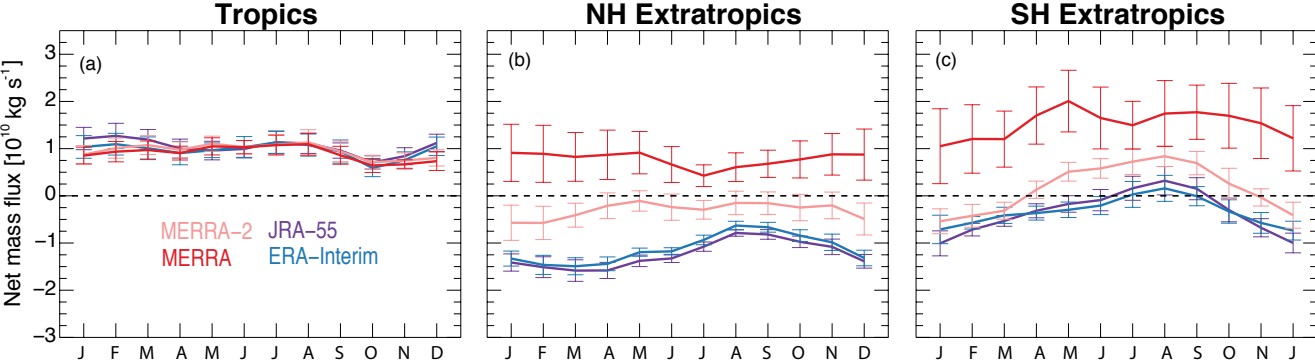

**Figure 12.** As in Figure 11, but for the (a) tropics, (b) NH extratropics, and (c) SH extratropics.

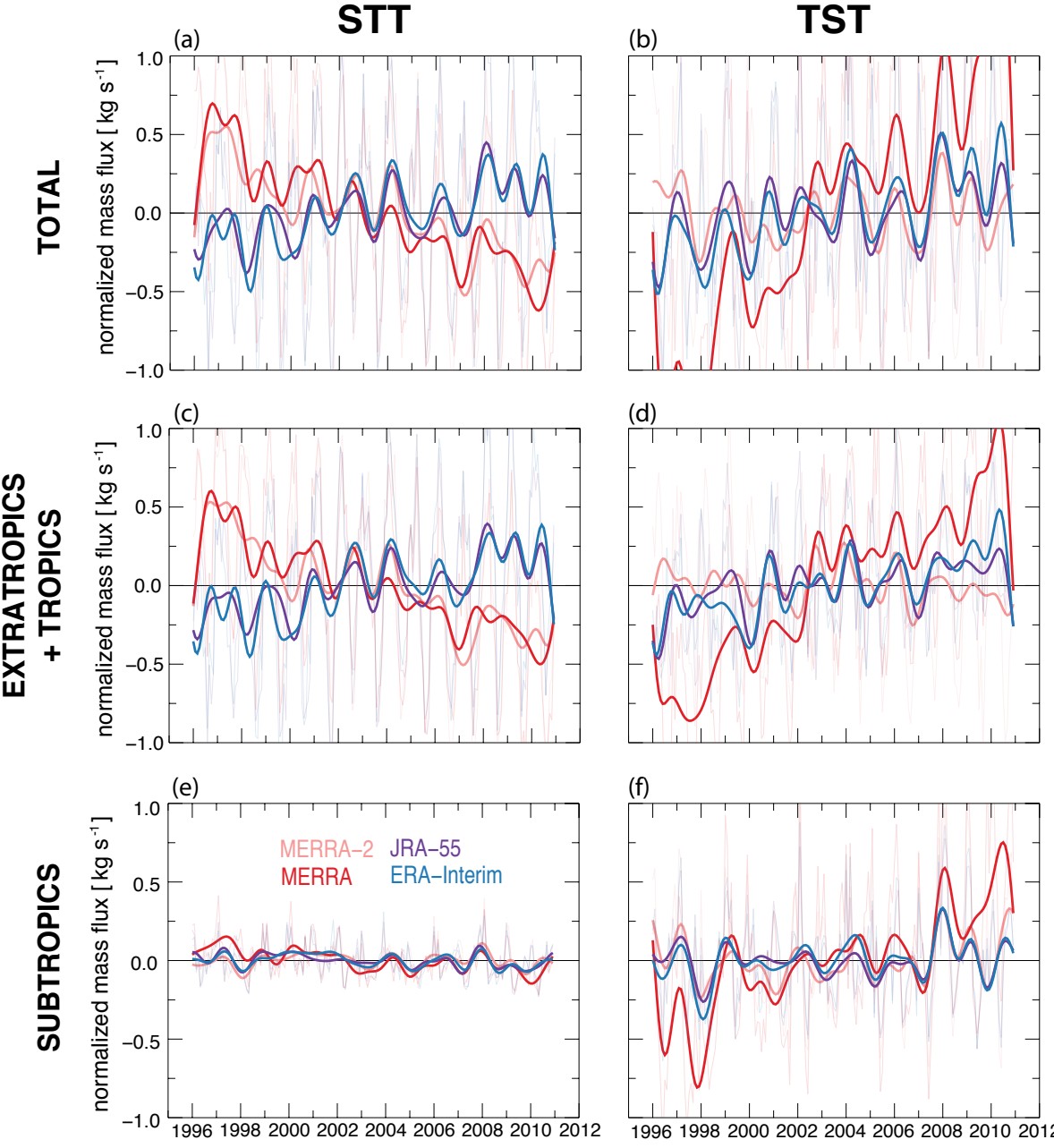

**Figure 13.** For each reanalysis, time series of globally integrated (top) total, (bottom) subtropical, and (middle) combined extratropical and tropical STT (left) and TST (right) mass fluxes that are with respect to the 15-year study period mean (1996–2010). The thin lines represent the monthly mean mass fluxes, while the bold lines are the result of applying a high-pass filter to a Fourier transform of each time series (power at time scales ≤ 12 months is removed). STE from JRA-55 is shown by the purple lines, ERA-Interim by the blue lines, MERRA-2 by the light red lines, and MERRA by the dark red lines.

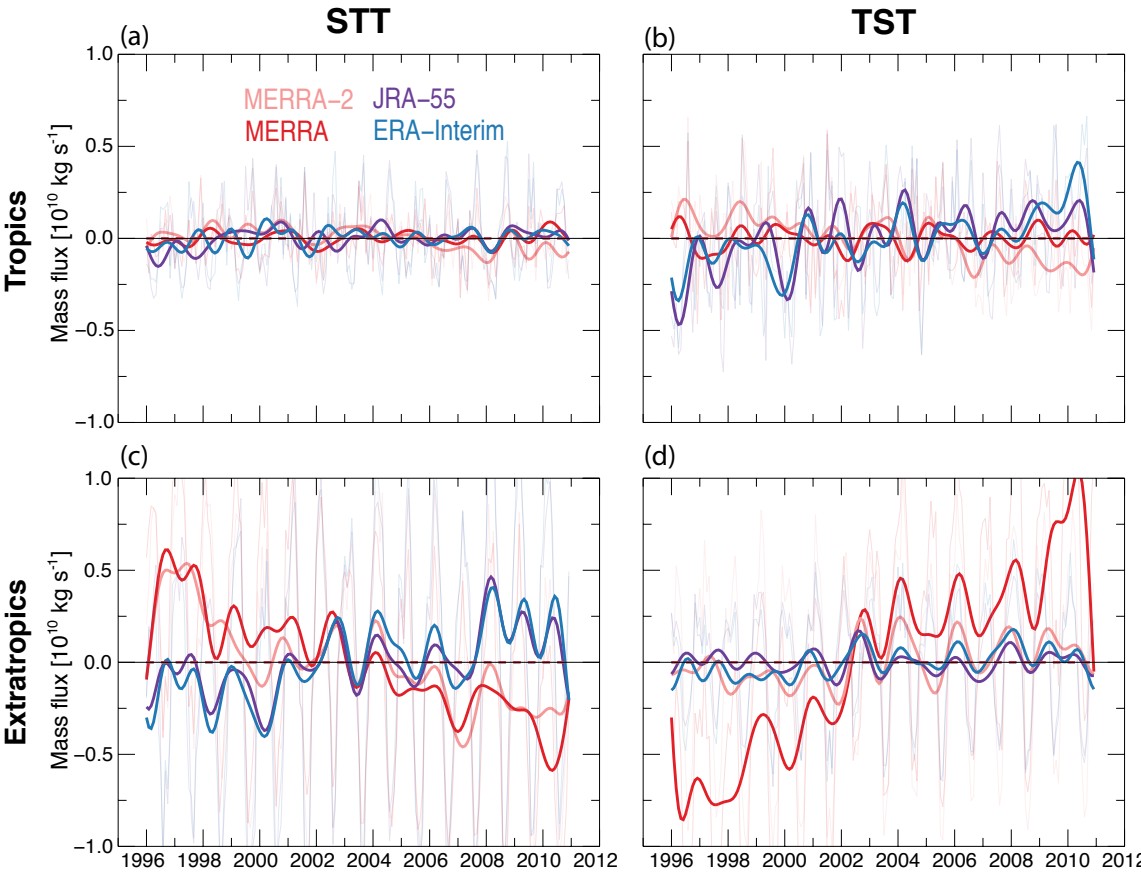

**Figure 14.** As in Fig. 13, but for STE in the (top) tropics and (bottom) extratropics.

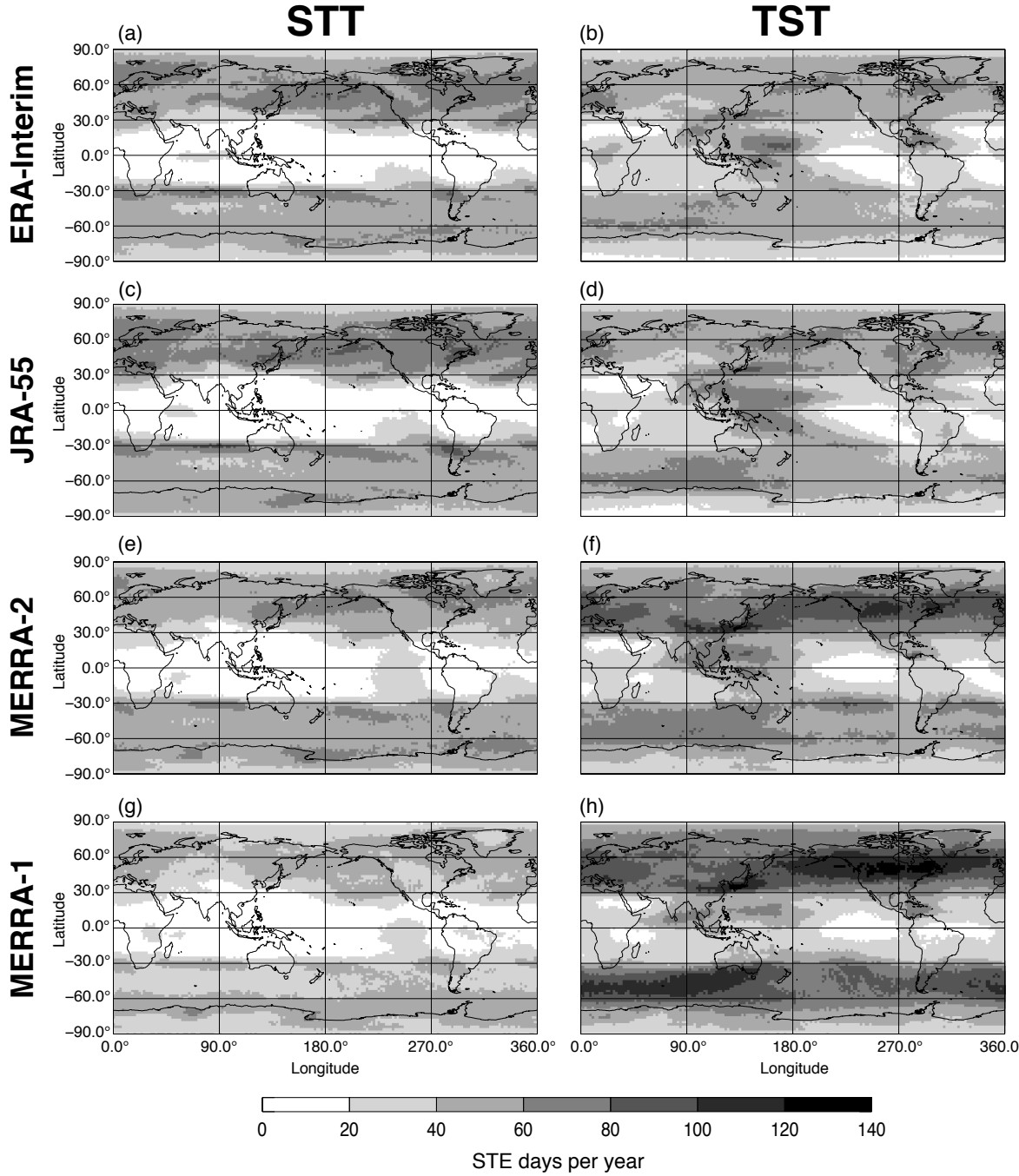

**Figure 15.** Global occurrence frequency distributions of (left) STT and (right) TST events for (a & b) ERA-Interim, (c & d) JRA-55, (e & f) MERRA-2, and (g & h) MERRA.

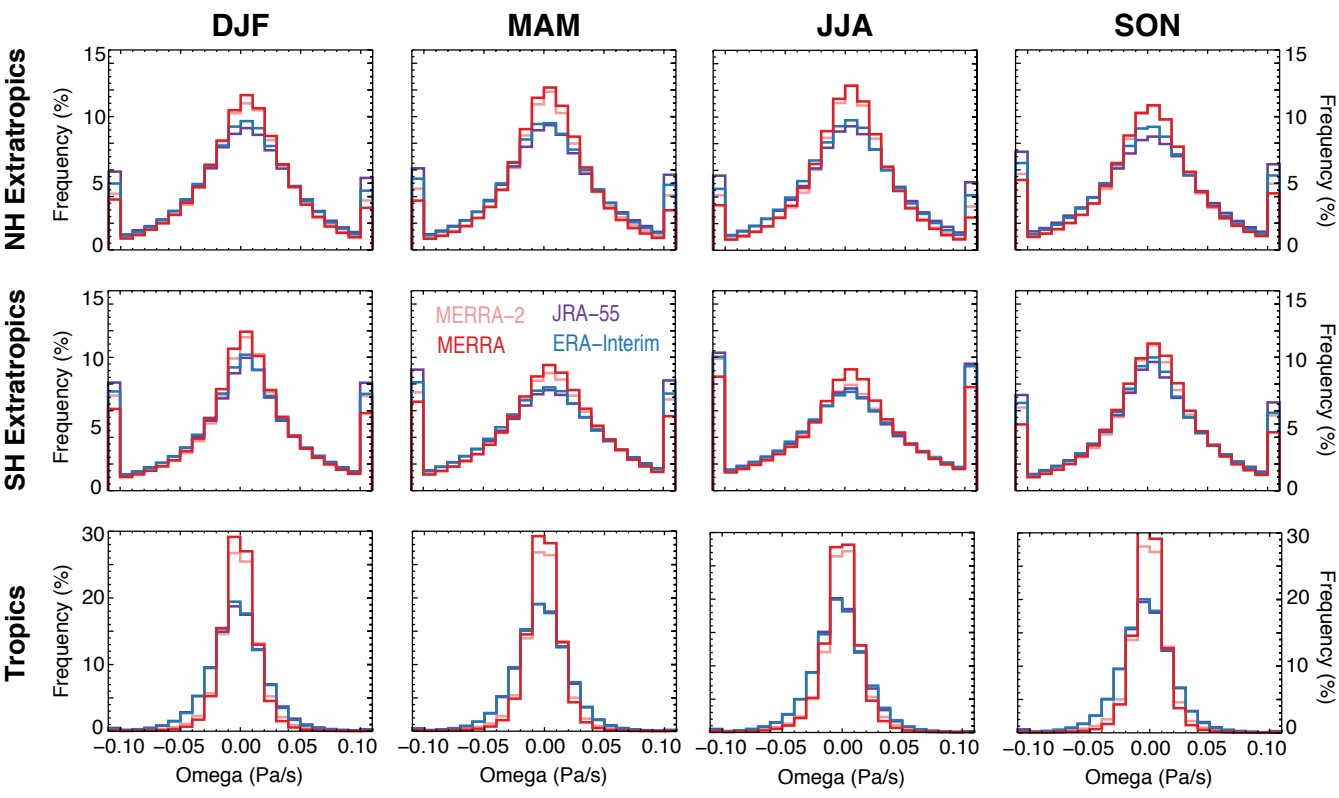

**Figure 16.** Probability density functions (PDFs) of vertical wind (Omega; Pa/s) at the tropopause in the (top row) NH extratropics, (middle row) SH extratropics, and (bottom row) tropics for each season (columns from left to right are DJF, MAM, JJA, and SON, respectively). JRA-55 is shown by the purple lines, ERA-Interim by the blue lines, MERRA-2 by the light red lines, and MERRA by the dark red lines. Note: the vertical wind extremes (i.e., those beyond the limits of the abscissa) are consolidated into the leftmost and rightmost bins of each PDF.

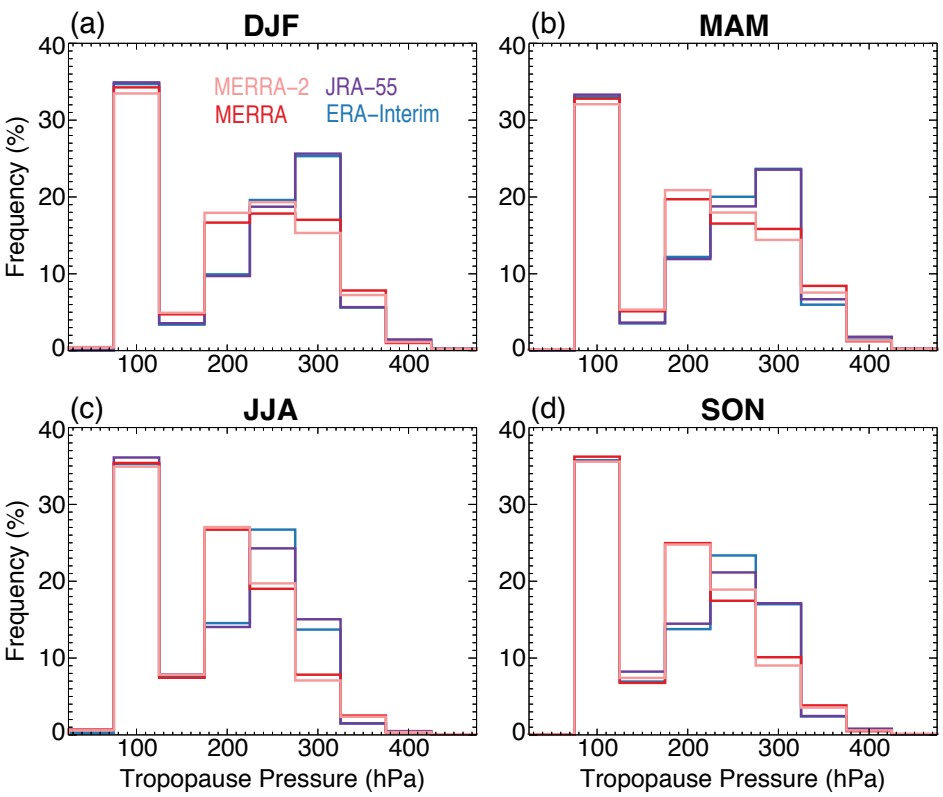

**Figure 17.** Probability density functions (PDFs) of global tropopause pressure (hPa) over the 15-year period (1996-2010), separated by season: (a) DJF, (b) MAM, (c) JJA, and (d) SON. JRA-55 is shown by the purple lines, ERA-Interim by the blue lines, MERRA-2 by the light red lines, and MERRA by the dark red lines.