# Peer review of "Global large-scale stratosphere-troposphere exchange in modern reanalyses"

_Atmospheric Chemistry and Physics, 2016_

## Referee Comment (RC1) · Anonymous Referee #2 · 23 Nov 2016

The paper presents a climatology of stratosphere-troposphere exchange (STE) in four modern reanalyses. The results show coherence in the overall spatial structure but also reveal interesting differences among the reanalyses. The study is based on the lapse rate tropopause, which provides a picture of UTLS transport more consistent with the stratospheric global overturning circulation as compared to previous works using a PV contour. Perhaps the most relevant difference is that over the extratropics troposphere-to-stratosphere transport is larger than stratosphere-to-troposphere in the MERRA reanalyses, while in the other two downward transport dominates. The analyses presented are comprehensive and novel, and the paper is well written and generally easy to follow. This work contributes to advance in understanding the

[Figure]

UTLS transport processes, and I recommend its publication after the following minor comments and suggestions have been addressed.

*General comments*

- P6 L19-20: Since you include tropical upwelling as part of the TST, I would suggest including extratropical downwelling as part of the STT. You actually include it in the classification in Section 4.5 when referring to trends in the time series, so why not explicitly include it from the beginning?

- P10 L19: In the analyses of the annual cycle, I would suggest separating three regions: tropics, NH extratropics, SH extratropics. As it is, the annual cycle is mixing several different mechanisms and perhaps this alternative separation would provide clearer insights on the causes of the differences among reanalyses.

*Specific comments*

- P12 L21-26: How do the results in this study compare to previous studies explicitly looking at the BDC in reanalyses (Abalos et al. 2015 JRG-A, Miyazaki et al. 2016, ACP)?

- Section 4.6.2. Diagnostics: I am missing some discussion at the end of the Section connecting these diagnostics to the previous results shown in the paper. For instance, are the differences in tropopause height and/or jet location consistent with more extratropical TST in MERRA?

- Section 5.2. Discussion: Please add some discussion on how your estimates compare quantitatively with previous works highlighted in the Introduction.

*Technical corrections/suggestions*

- P1 L15: "has important and significant": perhaps redundant?
- P1 L20: although water vapor is a greenhouse gas, is it considered a pollutant?
- P5 L13: "pvu" should be PVU
- P5 L16: "about 750 m in the extratropical UTLS to about 1100 m in the tropical UTLS". You could specify an approximate range of altitudes corresponding to the extratropical and tropical UTLS
- P5 L23: "6 billion". It is better to write $6 \cdot 10^9$, to avoid confusion with the world billion
- P6 L4: "affects" should be effects
- P8 L17: "East pacific" should be West Pacific, right?
- P9L31: "... STE mass flux" I suggest adding the clarification (TST-dominant)
- P11 L28: "normalize" usually refers to dividing by the time mean, while what you did was compute the anomalies with respect to the mean
- P12 L17: suggestion for clarity: " ... show increasing vertical STT in the extratropics and TST in the tropics, whereas ..."
- P15 L27: suggestion: remove "there to be"
- P15 L29-30: sentence not clear, rephrase.

---

## Referee Comment (RC2) · Anonymous Referee #1 · 15 Dec 2016

The authors perform an analysis of stratosphere troposphere exchange (STE) in four state-of the art reanalysis data sets (ERA Interim, MERRA, MERRA2, JRA 55). They apply Lagrangian analysis to diagnose exchange using the thermal tropopause as reference surface. STE is subdivided according to spatial directions of exchange 'lateral' and 'vertical'. Before starting the reanalysis comparison, they compare their method with the results of a recent analysis of STE from Skerlak et al., who used a PV-based threshold. They find differences, which are based on the different methods and motivate their own lapse-rate-tropopause (LRT) approach partly from these different results.

The reanalysis data are analysed for time period of 15 years, despite longer analysis

time periods would have been possible. They find substantial differences between STE in the reanalysis data sets. Whereas JRA and ERA Interim are STT dominated, MERRA 1 and 2 are TST dominated, according to the authors. Mass fluxes are shown and exhibit significant differences and net mass fluxes partly deviate significantly from zero, which is mentioned, but not explained or discussed in detail.

The manuscript could in principle make an important contribution to the field, since a consistent comparison of exchange between different reanalysis data is of very high interest. However, the authors need to assess the caveats of their method and the consequences for the result. It seems to me that they miss a part of the processes particularly at the extratropical tropopause. This might be due to the method and procedure they have applied to diagnose STE. The non-zero net fluxes also could be an indication for this.

Further a careful quantification and discussion of the tropopause location and its determination is essential for the paper and needs to be included. Based on this and on the points below, the authors should discuss the results, which are of high importance and interest more carefully also in the light of the potential caveats of their methods of tropopause determination or differences in the tropopause location between reanalysis data sets.

1) The thermal tropopause itself needs to be assessed for the individual data sets, before analysing the exchange and probably before regridding (see also suggestions below). This point is crucial, particularly for the method as applied here. Which role plays the interpolation of the fields for the results, particular for the vertical coordinate and the location of the tropopause altitude?

2) The authors just perform a spatial classification of STE 'lateral' and 'vertical', which does not mirror the dynamical processes. For exchange between the subtropics and mid latitudes, where the tropopause break has a large vertical extent, this might work well. For the mid latitudes they might miss parts of the exchange (see comments below

with references) since there is no 'lateral' STE per definition of the method. This needs to be discussed as well and potentially lead to a potential bias e.g. in the fluxes.

I highly suggest to include the method of Skerlak et al., 2014, despite differences, since it allows for a further independent comparison also with previous results from literature.

3) As stated by the authors, one should expect the STE being mass conservative. This seems however not be the case. Since this is a central point also for the long-term STE time series the authors should also discuss carefully the caveats of their method.

4) The thermal tropopause definition in general, but especially in high latitudes is problematic, how does this affect the results (see e.g. Zangl and Hoinka, 2000)?

5) How do the results relate to other approaches?

Overall such a comparison is a valuable effort. However, in the current form the paper needs a major revision, particularly with regard to potential uncertainties of the method, which could contribute to the discrepancies between the reanalysis data sets.

MAJOR: Which role plays interpolation of the fileds for the results? Did the authors interpolate also in the vertical? If yes I think a sensitivity for at least one model should be done to assess the effect of interpolation ob the STE results.

Further the authors find the largest differences between the data sets for the 'vertical' exchange. This is not surprising, since it might be related to differences in the vertical resolution or the variability of the vertical wind in the models. Also the differences in the representation of the thermal tropopause might contribute to these differences, which in turn depends on the vertical resolution of the specific data set. I missed an assessment of this particularly for the extratropics (e.g. a monthly pdf of vertical wind for each month the extratropics).

Since the spatial coordinates play such a crucial role the authors need to systematically assess this: They should add plots (PDFs) of the tropopause height separated for the extratropics (seasonally resolved) and tropics for each data set. This should be done

for the original data as well as for the interpolated data to get both, the differences between the data sets and the effect of regridding.

For STE: Evaluating the differences TP_tropical_press minus TP_extratropical_press between the different data sets is important since differences of the diagnosed separation between extratropical and tropical tropopause will directly affect STE results (see also comments further below).

Criteria: 'Lateral' STT: Is exchange across the extratropical tropopause possible, which is not 'vertical'? How are particles counted, which start in the troposphere, but follow downward sloping isentropes into the stratosphere? These parcels (initially lying below the extratropical TP) descend e.g. from above the polar jet and are mixed into the adjacent stratosphere above a trough). Such a parcel will descend, but gain PV? This is not an exotic process and does occur quite frequent (e.g. Pan et al., 2007, Pan and Konopka., 2012, see also Juckes, 2000). How is quasi-isentropic mixing in the extratropics in general treated? According to the classification no 'lateral' exchange is possible if the tropopause is below 200 hPa (i.e. at higher pressures). Also: Why does exchange above and below the jets need to be 'vertical' (p.6, l.33)? This is a limitation of the method and needs to be clearly discussed, also in comparison to Skerlak et al., 2014.

SPECIFIC: p.5, l.10-12: It has been shown in many studies (Gettelman et al., and references therein) that in the extratropics away from the subtropical jet very well represents tracer isopleths. This is due to the fact that PV is materially conserved under adiabatic conditions, which is not the case for the LRT. Notably the vertical gradient is included in the PV definition, which therefore inherently includes the thermal definition. Note further, that the -2K/km are an arbitrary definition for the thermal tropopause gradient in a similar way as a fixed PV threshold. Therefore, the above argument is not valid.

p.6, l.19: Isn't convection 'vertical'? How is it considered? p-7, l.15: What is 'systematic' upwelling? p.7, l.17: 'according to our knowledge... LRT method agree more closely

with known transport mechanisms'. Please give a few references here, such a statement without references is very superficial. How does this compare with e.g. Juckes et al., 2000? Maybe the LRT method is very well suited for identifying exchange at location of the subtropical jet. At mid latitudes the the LRT altitude criteria probably fail in regions?

p.13,l.20: What is 'equivalent dynamics'? p.13,l.13-l.22; The arguments are confusing as well as the use of 'dynamical and physical differences': Why is the jet location a dynamical difference, the fold and tropopause physical differences? Both are related to the same physical processes, which control temperature gradients and pressure etc. and finally the location of these structures - based on the representation of physics in the respective reanalysis model.

Further: Why can TP altitude lead to changes in STE between the different models? If the tropopause location (and jet, folds) in each reanalysis is the result of differences in the respective model physics, it still might be self consistent within each reanalysis data set. One could get differences of tropopause height, location, jets etc. between different data sets without differences in STE.

Since 'vertical' exchange is so important - which role plays the variability of the vertical wind in the data sets?

p.14, l.13/13.: Vertical and lateral are inappropriate terms to characterize physical processes, It's just a spatial direction in (cartesian) coordinates, but STE is more complex as you show. Therefore please change the word 'processes' to 'direction'

p.14, l.25-27: This statement as it stands here is incorrect or at least misleading. In geometrical coordinates the direction could be downward, although the PV change can be positive. This were a TST in the physical víew accounting for thermodynamics, but an STT from geometrical aspects.

p.14, l.30: How is transport 'stratified'?

p.15, l.12: Why is poleward transport the same as TST?

References:

Juckes, M.: The descent of tropospheric air into the stratosphere, QJR, 2000, 10.1002/qj.49712656216

Pan and Konopka, On the mixing driven formation of the ExTL, JGR 2012, 10.1029/2012JD017876. (Fig,1)

Pan et al., Chemical behavior of the tropopause observed during the Stratosphere-Troposphere Analyses of Regional Transport experiment, JGR, 2007, 10.1029/2007JD008645 (Fig.8)

Zängl and Hoinka, The Tropopause in polar regions, JAS, 2000.

---

## Author Comment (AC1) · 3 Feb 2017

Response to Anonymous Referee #2

General comment #1:
- P6 L19-20: Since you include tropical upwelling as part of the TST, I would suggest including extratropical downwelling as part of the STT. You actually include it in the classification in Section 4.5 when referring to trends in the time series, so why not explicitly include it from the beginning?

> P6 L31-33 now reads: "Exchanges in the extratropics and tropics, however, are associated with other processes, such as extratropical cyclones, stratospheric intrusions, downwelling in the extratropics and upwelling in the tropics."

General comment #2:
- P10 L19: In the analyses of the annual cycle, I would suggest separating three regions: tropics, NH extratropics, SH extratropics. As it is, the annual cycle is mixing several different mechanisms and perhaps this alternative separation would provide clearer insights on the causes of the differences among reanalyses.

> Based upon the suggestion, we adjusted the annual cycles as recommended and found that the differences in NH TST were eliminated (updated Figure 9). Furthermore, the tropical annual cycles (updated Figure 10) reveal more separation between the reanalyses and show two preferred modes of TST: weakly bimodal for JRA-55 and ERA-Interim and unimodal for both MERRA reanalyses. These changes take place from P10 L25 to P12 L9 in the revised manuscript. The revised figures are provided below.

> Figure 9:

[Figure]

Figure 10:

[Figure]

Specific comment #1:
- P12 L21-26: How do the results in this study compare to previous studies explicitly looking at the BDC in reanalyses (Abalos et al. 2015 JGR-A, Miyazaki et al. 2016, ACP)?

    P13 L4-11 now reads: "Changes in the speed of the BDC have been examined in previous studies. For example, Abalos et al. (2015) evaluated the dynamics of the BDC using ERA-Interim, JRA-55, and MERRA and show that there is general agreement in a strengthening BDC over the period 1979-2012 by 2-5% per decade. Observational studies show decreases in tropical stratospheric water vapor, ozone, and temperature observed by satellite, which also corresponds to an increase in tropical upwelling associated with an accelerated BDC (Randel et al., 2006). Chemistry- climate models have also indicated an acceleration of the BDC over time (e.g., Austin and Li, 2006). These previous reanalysis, observational, and modeling studies are consistent with the results from ERA-Interim and JRA-55 here, while MERRA-2 is in disagreement and MERRA does not indicate changes in the BDC over time in our analysis."

Specific comment #2:
- Section 4.6.2 Diagnostics: I am missing some discussion at the end of the section connected these diagnostics to the previous results shown in the paper. For instance, are the differences in tropopause height and/or jet location consistent with more extratropical TST in MERRA?

    We have significantly altered section 4.6.2 in the revision and attempted to better connect the diagnostics with STE results.

Specific comment #3:
- Section 5.2 Discussion: Please add some discussion on how your estimates compare quantitatively with previous works highlighted in the introduction.

    P16 L21-29 now reads: "Third, as referred to in the Introduction, large quantitative uncertainties in STE exist from previous Eulerian and

Lagrangian STE studies. In particular, estimates for STE have often been limited to specific regions or time periods or based on inadequate and/or incomplete methods (compared to that possible with current methods and computational abilities). Here, we found that mean net STE magnitudes also range considerably when an equivalent method is applied to multiple modern reanalyses (e.g., see Table 1). However, few previous studies enable direct comparison with our estimates. In particular, the alternative PV-based approach by Škerlak et al. (2014) is arguably the most direct, where ERA-Interim net STE integrated globally over the 15 yr period in our study is approximately $1.48 \times 10^{17}$ kg/yr downward (STT), while it is about 3.5 times smaller ($4.2 \times 10^{16}$ kg/yr) in the Škerlak et al. (2014) study. These differences do not necessarily suggest that one method is superior to the other, but that two largely similar Lagrangian approaches can yield substantially different results due to the employed troposphere-stratosphere boundary (e.g., see Figs. 2 & 3)."

Technical comments/suggestions and responses:

- P1 L15: "has important and significant": perhaps redundant?
  This has been changed to "has significant"

- P1 L20: although water vapor is a greenhouse gas, is it considered a pollutant?
  Thank you for pointing out this misleading remark. This has been revised to "water vapor and tropospheric pollutants, such as carbon monoxide, …"

- P5 L13: "pvu" should be PVU
  Corrected.

- P5 L16: "about 750 m in the extratropical UTLS to about 1100 m in the tropical UTLS". You could specify an approximate range of altitudes corresponding to the extratropical and tropical UTLS.
  We have now specified approximate altitude ranges at P5 L27.

- P5 L23: "6 billion". It is better to write $6 \bullet 10^9$, to avoid confusion with the word billion.
  Done.

- P6 L4: "affects" should be effects.
  Done.

- P8 L17: "East Pacific should be West Pacific, right?
  Yes, this has been corrected.

- P9 L31: "…STE mass flux" I suggest adding (TST-dominant)
  Done.

- P11 L28: "normalize" usually refers to dividing by the time mean, while what you did was compute the anomalies with respect to the mean.
    - P12 L13-14 now reads: "In Fig. 13, global time series of STT anomalies and TST anomalies are shown with respect to their mean mass fluxes (i.e., 15-yr means are removed)."

- P12 L17: suggestion for clarity: "… show increasing vertical STT in the extratropics and TST in the tropics, whereas…"
    - Done.

- P15 L27: suggestion: remove "there to be"
    - Done.

- P15 L29-30: Sentence not clear, rephrase
    - P16 L8-9 now reads: "Taken together, these physical and dynamical differences may be significant sources of variability for climatological analyses and it is likely that they contribute to some of the STE differences observed in this study."

---

## Author Comment (AC2) · 3 Feb 2017

Response to Anonymous Referee #1

1) The thermal tropopause itself needs to be assessed for the individual data sets, before analyzing the exchange and probably before regridding (see also suggestions below.) This point is crucial, particularly for the method as applied here. Which role plays the interpolation of the fields for the results, particular for the vertical coordinate and the location of the tropopause altitude?

> We now include a comparison of the reanalysis lapse-rate tropopause altitudes in the diagnostics section (see additional response below). With regard to interpolation, we only interpolate in the horizontal dimension prior to applying the WMO algorithm, so the effect of interpolation on the altitude of the tropopause is negligible (see figure here using 6-hr ERA-Interim output for an entire month). This point has been made clear in the revised manuscript at P4 L20-26.

[Figure]

2) The authors just perform a spatial classification of STE 'lateral' and 'vertical', which does not mirror the dynamical processes. For exchange between the subtropics and mid latitudes, where the tropopause a large vertical extent, this might work well. For the mid latitudes they might miss parts of the exchange (see comments below with references) since there is no 'lateral' STE per definition of the method. This needs to be discussed as well and potentially lead to a bias e.g. in the fluxes.
I highly suggest including the method of Škerlak et al., 2014, despite differences, since it allows for a further independent comparison also with previous results from literature.

> Thank you for these comments. We have recognized following the reviews that our labeling of STE as "vertical" and "lateral" was somewhat misleading. All exchanges across the lapse-rate tropopause are considered in our trajectory analysis if they uphold the residence time and potential vorticity difference criteria; it is not dependent on the geometry of transport (save for the tropopause-relative pressure check, which is a local condition that does not require the exchange to have taken place in the

vertical dimension). We have re-named our transport classifications in the revised manuscript to reflect this issue and acknowledge the more appropriate regional distinction. The new classifications are "tropical", "subtropical" (previously lateral), and "extratropical". Since the previous use of "vertical" was meant to provide distinction from our so-called "lateral" exchanges across the tropopause break, its use has been removed from the revised manuscript (except for the general descriptions of individual transport processes in the Introduction).

Also, it is not clear what aspects of the Škerlak et al approach are desired/requested here. We do include a substantial comparison between their STE estimates and ours (i.e., Figs. 2 & 3 and attendant discussion), but note that beyond the use of a 3-D PV labeling methods, our Lagrangian STE approach and theirs is nearly equivalent. The only major difference is the troposphere-stratosphere boundary employed. While we appreciate the labeling method used in Škerlak et al., 2014 (and previous work), we believe it is somewhat impractical given our use of the lapse-rate tropopause in our study and provides unnecessary complexity for the overarching goal of this work: comparison of STE in multiple reanalyses.

3) As stated by the authors, one should expect the STE begin mass conservative. This seems however not be the case. Since this is a central point also for the long-term STE time series the authors should also discuss carefully the caveats of their method.

Yes, we do set an expectation that net STE estimates should be near zero as a result of mass conservation. As our method only counts air parcels that are "irreversibly" exchanged, we do miss many transient exchanges along the LRT (which may occur preferentially in one direction). We have made an effort to ensure that the caveats associated with our analysis are outlined well in the revised mansucript. Also, it is important to note that PV-based methods (such as Škerlak et al., 2014) show an imbalance in STE mass flux similar to that in our analysis.

4) The thermal tropopause in general, but especially in high latitudes is problematic, how does this affect the results (see e.g. Zangl and Hoinka, 2000)?

We disagree that the lapse-rate tropopause is generally problematic, but do acknowledge that there are issues with its use in the Antarctic polar region (south of 60S) during SH winter, as outlined in Zangl and Hoinka, 2001. We have mentioned this potential issue in the paper, but note that it is limited to ~3 months per year and about 12% of the area of the Southern Hemisphere. Regardless of its limitations in that sense, hemispheric STE fluxes show similar seasonal variability when comparing our lapse-rate tropopause based methods and PV-based methods.

5) How do the results relate to other approaches?

Only Lagrangian studies of STE (very few exist) can be used to compare spatial distributions, which are consistent with our results, aside from those outlined in the detailed comparison with Škerlak et al., 2014 included in the paper (Section 3). Our discussion (updated section 5.2) now includes more quantitative comparison of globally integrated net mass fluxes between our approach and previous ones (see also response to Reviewer 2).

MAJOR: Which role plays interpolation of the fields for the results? Did the authors interpolate also in the vertical? If yes I think a sensitivity for at least one model should be done to assess the effect of interpolation of the STE results.

As outlined above in response to a similar comment, interpolation to a coarser grid was only performed in the horizontal (which was made clear in the revised manuscript). As for its impact on the result of the trajectory calculations to determine STE, there have been many studies in the past that examine this problem (please see the Stohl, 1998 review on this topic). At wind field temporal resolution similar to that available in this study (6 hr), the effects of spatial resolution on the trajectory result are minimal. Temporal resolution (which we cannot change) is considerably more important for reducing errors in large-scale trajectories. Horizontal resolution is more important when the temporal resolution is sufficiently fine (< 4-6 hr). We have included these details on P4 L20-26 in the revised manuscript.

References:
Stohl, A., 1998: COMPUTATION, ACCURACY AND APPLICATIONS OF TRAJECTORIES -- A REVIEW AND BIBLIOGRAPHY, *Atmos. Environ.*, **31** (6), 947-966.

Further the authors find the largest differences between the data sets for the 'vertical' exchange. This is not surprising, since it might be related to differences in the vertical resolution or the variability of the vertical wind in the models. Also the differences in the representation of the thermal tropopause might contribute to these differences, which in turn depends on the vertical resolution of the specific data set. I missed an assessment of this particularly for the extratropics (e.g. a monthly pdf of vertical wind for each month the extratropics)

We have rewritten the diagnostics section in the paper and provided PDFs of vertical motion (separated by region and season) and tropopause pressure (separated by season). These figures are also included here. The differences in the reanalyses are clearer from these comparisons and show consistency among the models with similar vertical grids (as seen in STE). Vertical wind PDFs of MERRA and MERRA-2 should reveal higher frequencies of negative omega (ascent) at the tropopause as a result of TST-dominant exchange in the extratropics, however the TST-Dominant reanalyses do not show a positive skew, rather the shape of the PDFs are

similar throughout the annual cycle among each reanalysis. More information can be drawn from the tropics. PDFs show a slight positive skew indicating a larger frequency of ascent at the tropopause in the tropics. Moreover, the vertical winds are weaker in both MERRA and MERRA-2 compared to ERA-Interim and JRA-55 (STT-dominant).

[Figure]

Since the spatial coordinates play such a crucial role the authors need to systematically assess this: They should add plots (PDFs) of the tropopause height separated for the extratropics (seasonally resolved) and tropics for each data set. This should be done for the original data as well as for the interpolated data to get both, the differences between the data sets and the effect of regridding.

See previous comments.

For STE: Evaluating the differences TP_tropical_press minus TP_extratropical_press between the different data sets is important since differences of the diagnosed separation between extratropical and tropical tropopause will directly affect STE results (see also comments further below).

PDFs of tropopause pressure from the reanalyses were shown above. MERRA and MERRA-2 reveal a shallower transition from tropical to extratropical tropopause pressures in each season. A frequency minimum between tropical and extratropical tropopause modes of 150 hPa is shown to occur in each season for each reanalysis (as outlined previously in the manuscript). Tropopause altitudes are frequently higher (i.e., lower pressures) in the extratropics for the MERRA and MERRA-2 reanalyses. Tropopause pressures are largely consistent in the tropics, with differences between the reanalyses mostly due to slight offsets in the location of vertical model levels (not apparent here, but note that additional resolution in the PDFs is not appropriate given the native resolution of each reanalysis).

Criteria: 'Lateral' STT: Is exchange across the extratropical tropopause possible, which is not 'vertical'? How are particles counted, which start in the troposphere, but follow downward sloping isentropes into the stratosphere? These parcels (initially lying below the extratropical TP) descend (e.g. from above the polar jet and are mixed into the adjacent stratosphere above a trough). Such a parcel will descend, but not gain PV? This is not an exotic process and does occur quite frequent (e.g. Pan et al., 2007, Pan and Konopka, 2012, see also Juckes, 2000). How is quasi-isentropic mixing in the extratropics treated? According to the classification no 'lateral' exchange is possible if the tropopause is below 200 hPa (i.e. at higher pressures). Also: Why does exchange above and below the jets need to be 'vertical' (p.6, L.33)? This is a limitation of the method and needs to be clearly discussed, also in comparison to Škerlak et al., 2014.

As outlined above, the use of "vertical" and "lateral" was misleading. Since these are three-dimensional trajectories, any geometric evolution can result in STE if the tropopause-relative pressure changes during a parcel's transit through the atmosphere. What matters for exchange is the passing of a local condition (i.e., a parcel must lie above/below the local tropopause at its initial time and below/above the local tropopause at its final time). This could result from transport that is mostly vertical in nature, horizontal in nature and, possibly, that which deviates from the conceptual framework of upward = TST and downward = STT. We have

revised the text where necessary to avoid this confusion, including the renaming of our STE classifications (as outlined above).

SPECIFIC: p.5, L.10-12: It has been shown in many studies (Gettelman et al., and references therein) that in the extratropics away from the subtropical jet very well represents tracer isopleths. This is due to the fact that PV is materially conserved under adiabatic conditions, which is not the case for the LRT. Notably the vertical gradient is included in the PV definition, which therefore inherently includes the thermal definition. Note further, that the -2K/km are an arbitrary definition for the thermal tropopause gradient in a similar way as a fixed PV threshold. Therefore, the above argument is not valid.

This text has been modified a bit here. The intended message is that the advantage of using the lapse-rate tropopause is its ability to characterize the tropopause globally and its common coincidence with the largest chemical gradients between troposphere and stratosphere, while the use of a fixed PV threshold/gradient requires a second definition for the tropics (a limitation this paper seeks to avoid).

p.6, L.19: Isn't convection 'vertical'? How is it considered?

Convection is not a resolved process in the reanalyses due to the coarse nature of their native grid resolutions, and therefore not considered in this study, which was previously mentioned at the end of the Introduction.

p.7, L.15: What is 'systematic' upwelling?

Replaced with "ubiquitous".

p.7, L.17: 'according to our knowledge…LRT method agree more closely with known transport mechanisms'. Please give a few references here, such a statement without references is very superficial. How does this compare with e.g. Juckes et al., 2000? Maybe the LRT method is very well suited for identifying exchange at location of the subtropical jet. At mid latitudes the LRT altitude criteria probably fail in regions?

Done.

p.13, L.20: What is 'equivalent dynamics'?

This is no longer relevant given the significant revisions to the diagnostics section in the paper.

p.13, L.13-22: The arguments are confusing as well as the use of 'dynamical and physical differences': Why is the jet location a dynamical difference, the fold and tropopause physical differences? Both are related to the same physical processes, which control temperature gradients and pressure etc. and finally the location of these structures – based on the representation of physics in the respective reanalysis model.

See previous.

Further: Why can TP altitude lead to changes in STE between the different models? If the tropopause location (and jet, folds) in each reanalysis is the result of differences in the respective model physics, it still might be self consistent within each reanalysis data set. One could get differences of tropopause height, location, jets etc. between different data sets without differences in STE.

See previous.

Since 'vertical' exchange is so important – which role plays the variability of the wind in the data sets?

See previous responses to similar comments.

p.14, L.13: Vertical and lateral are inappropriate terms to characterize physical processes, it's just a spatial direction in (Cartesian) coordinates, but STE is more complex as you show. Therefore please change the 'processes' to 'direction'.

See previous.

p.14, L.25-27: This statement as it stands here is incorrect or at least misleading. In geometrical coordinates the direction could be downward, although the PV change can be positive. This were a TST in the physical view accounting for thermodynamics, but an STT from geometrical aspects.

This has been revised in line with changes outlined in response to previous comments.

p.14, L.30: How is transport 'stratified'?

Replaced with "separated".

p.15, L.12: Why is poleward transport the same as TST?

This point has been clarified. This refers to transport across the tropopause break, which is inherently poleward if TST and equatorward if STT.

---

## Author Response (AR2)

**21 March 2017**

Dear Editor Stiller,

We thank Reviewer 1 once again for their helpful comments on our manuscript. Reviewer comments, our responses (in blue), and a copy of the revised manuscript with corresponding revisions of the text italicized are provided below.

Sincerely,
Alex Boothe & Cameron Homeyer

Response to Anonymous Referee #1

The authors have included new figures showing winds and tropopause height differences between the models, which are really interesting and might hint toward the significant differences regarding TST - it would be great to have some potential hypotheses (e.g. at the end of 5.1) for potential mechanisms relating the TP height differences to TST.

> We have added a short statement on how the differences shown in Figures 15-17 might contribute to differences in STE and suggested a potential pathway for future research on this topic at the end of Section 5.2 (page 17, lines 12-17).

A second point is surprising: Looking at Figure 2 it seems that STT by SKERLAK 2014 is much stronger than calculated in the manuscript, but at the end it is stated, that the STT in Skerlak et al., 2014 is three times smaller - is this correct?

> No changes made. The statement referred to near the end of the manuscript (page 17, line 3 of the previous submission) is in reference to *net STE*, not STT. We do highlight the differences in STT and TST individually with Figures 2 & 3, but the discussion here is focused on the difference (TST – STT) which is referred to as the net STE.

Finally the statement on page 5, l.15 is still misleading. It is not true that, the LRT represents best the chemical transition ( and gradient) globally , see e.g. Thouret et al., 2007).

> We respectfully disagree that this statement is still misleading. Following our previous revision, we no longer use language such as 'best' in this description, but state that the LRT commonly coincides with the sharpest stability and chemical transitions between the troposphere and stratosphere and provide supporting references. The Thouret et al paper referenced in this comment uses a PV-based tropopause definition, but does not provide a comparison of trace gas profiles relative to multiple tropopause definitions. The papers referenced in our manuscript do show comparisons of trace gases in relative altitude to multiple tropopause definitions, with noticeably sharper transitions for profiles relative to the LRT compared to those relative to a PV-based tropopause. Thus, we believe our description here is appropriate and this text remains unchanged.

[revised manuscript text omitted]